# HLTF disrupts Cas9-DNA post-cleavage complexes to allow DNA break processing

Giordano Reginato [1], Maria Rosaria Dello Stritto [1], Yanbo Wang[2,10], Jingzhou Hao [3,4], Raphael Pavani [5], Michael Schmitz [6], Swagata Halder[1,11], Vincent Morin[7], Elda Cannavo[1], Ilaria Ceppi [1], Stefan Braunshier[1], Ananya Acharya[1], Virginie Ropars [7], Jean-Baptiste Charbonnier[7], Martin Jinek [6], Andrè Nussenzweig [5], Taekjip Ha[2,3,4,8,9] & Petr Cejka [1] ✉

The outcome of CRISPR-Cas-mediated genome modifications is dependent on DNA double-strand break (DSB) processing and repair pathway choice. Homology-directed repair (HDR) of protein-blocked DSBs requires DNA end resection that is initiated by the endonuclease activity of the MRE11 complex. Using reconstituted reactions, we show that Cas9 breaks are unexpectedly not directly resectable by the MRE11 complex. In contrast, breaks catalyzed by Cas12a are readily processed. Cas9, unlike Cas12a, bridges the broken ends, preventing DSB detection and processing by MRE11. We demonstrate that Cas9 must be dislocated after DNA cleavage to allow DNA end resection and repair. Using single molecule and bulk biochemical assays, we next find that the HLTF translocase directly removes Cas9 from broken ends, which allows DSB processing by DNA end resection or non-homologous end-joining machineries. Mechanistically, the activity of HLTF requires its HIRAN domain and the release of the 3′-end generated by the cleavage of the non-target DNA strand by the Cas9 RuvC domain. Consequently, HLTF removes the H840A but not the D10A Cas9 nickase. The removal of Cas9 H840A by HLTF explains the different cellular impact of the two Cas9 nickase variants in human cells, with potential implications for gene editing.

The removal of protein blocks from DNA double-stranded breaks (DSBs) is catalyzed by the MRE11 nuclease acting within the conserved Mre11-Rad50-Xrs2 (MRX) complex in yeast or MRE11-RAD50-NBS1 (MRN) in human cells[1,2]. MRX/N functions in conjunction with phosphorylated Sae2/CtIP, which mediates cell cycle-dependent control of resection and hence homology-directed repair (HDR)[3–6]. The MRE11 endonuclease cuts the 5′ DNA strand on the end-distal side of the protein block, and was previously shown to process DSBs bound not only by non-physiological streptavidin but also by the non-homologous end-joining (NHEJ) factors Ku-DNAPK (KU70-KU80 complex; DNA-dependent protein kinase, catalytic subunit), single-stranded DNA binding protein RPA, or nucleosomes[7–12]. The MRX/MRN

[1]Faculty of Biomedical Sciences, Institute for Research in Biomedicine, Università della Svizzera italiana (USI), 6500 Bellinzona, Switzerland. [2]Department of Biophysics & Biophysical Chemistry, Johns Hopkins University, Baltimore, MD 21205, USA. [3]Department of Biophysics, Johns Hopkins University, Baltimore MD21218, USA. [4]Program in Cellular and Molecular Medicine, Boston Children's Hospital, Boston, MA 02115, USA. [5]Laboratory of Genome Integrity, National Cancer Institute, NIH, Bethesda, MD, USA. [6]Department of Biochemistry, University of Zurich, Winterthurerstrasse 190, 8057 Zürich, Switzerland. [7]Université Paris-Saclay, CEA, CNRS, Institute for Integrative Biology of the Cell (I2BC), 91198 Gif-sur-Yvette, France. [8]Department of Pediatrics, Harvard Medical School, Boston, MA 02115, USA. [9]Howard Hughes Medical Institute, Boston, MA 02115, USA. [10]Present address: Department of Pathology, Stanford University School of Medicine, Stanford, CA, USA. [11]Present address: Biological Systems Engineering, Plaksha University, Mohali, Punjab 140306, India. ✉e-mail: petr.cejka@irb.usi.ch

complex is also required for the removal of SPO11 complexes from meiotic DNA breaks, and plays an important role in the removal of topoisomerase DNA cleavage complexes[13–20]. Subsequent exonucleolytic action of MRE11 and/or downstream long-range resection nucleases such as EXO1 or DNA2 then exposes the 3′ terminated ssDNA, which channels the repair into HDR pathways[5,21,22]. How the endonuclease activity of MRE11 is controlled in order to prevent spurious incisions of the genome away from DSBs is not understood.

In the last decade, CRISPR-associated nucleases Cas9 and Cas12a revolutionized the field of genome editing and provided flexible tools that have been employed for clinical application[23,24]. Despite the variety of ways in which Cas9 and its variants are used, including base and prime editing, transcriptional control, and epigenetic modification, one of its most common applications remains the site-specific induction of DSBs for gene knock-outs, mediated by NHEJ, or HDR-mediated gene editing applications[25–29]. Cas9 has two nuclease domains. The HNH nuclease domain cleaves the DNA strand that is complementary to the RNA guide, defined as the target strand, while the RuvC-like nuclease domain cleaves the opposite non-complementary, or non-target, strand[30]. The DNA breaks induced by Cas9 are known to remain bridged, which delays their detection and processing[31–34]. Indeed, Cas9-dependent breaks are repaired with slower kinetics than breaks induced by radiation or other site-specific nucleases[34–36]. The residency time of Cas9 at DSB sites was also shown to affect the outcome of repair[37,38]. Hence, the identification of factors influencing Cas9 stability at the DSBs is highly relevant for gene editing. Several processes have been described to dislocate Cas9 from the break sites to free up the ends for NHEJ or DNA degradation by model bacterial exonucleases. These processes include transcription, replication, and chromatin remodeling by the FACT complex[39–41]. As CRISPR-based genome editing functions independently of these processes, additional factors might control Cas9 residency time at the break sites.

Considering the ability of the MRE11 complex to repair blocked DSBs, the MRE11 complex is a prime candidate for the initial recognition and processing of Cas9-induced breaks. Indeed, MRE11 ChIP-Seq is used to map sites of DNA cleavage by Cas9 in mammalian cells with a high degree of sensitivity[42]. Unexpectedly, we report here that the MRE11 complex per se cannot sense DNA breaks generated by Cas9 in vitro, and demonstrate that the DNA end-bridging activity of Cas9 prevents their processing and repair. Our data suggest that Cas9 must be removed, or one of the ends needs to be dislocated, before DNA end resection and/or repair can take place. In contrast, Cas12a-mediated breaks are readily resected, consistent with the PAM-distal end being released by Cas12a after DNA cleavage[43,44]. We show that the replication fork remodeler HLTF is highly efficient in removing Cas9 from the break, allowing Cas9 to act repetitively to cleave multiple target DNA molecules. Cas9 removal by HLTF allows the processing of the ends by DNA end resection machinery or NHEJ in vitro. HLTF-mediated removal of Cas9 is dependent on its HIRAN domain and on its motor-driven translocase activity. Additionally, we show that, unlike replication and transcription, this mode of Cas9 removal is specific for the post-cleavage state of wild type or the H840A Cas9 nickase, which releases the 3′-end of the broken non-target DNA strand[31]. The removal of Cas9 H840A but not Cas9 D10A nickase by HLTF explains the different impact of the two Cas9 nickase variants in cells[37,45,46].

## Results

### Cas9-dependent breaks are invisible to DNA end resection enzymes
The MRE11 complex is a central responder to DSBs[6,47,48]. It removes protein blocks from DSBs by cleaving the 5′-terminated DNA strands endonucleolytically on the end-distal side of protein blocks[1,2,7,8,10,11]. The block removal activity of the MRE11 complex would seem to be an ideal candidate for the resection of CRISPR-Cas-dependent breaks, as

the Cas nucleases are known to remain bound to the broken DNA after cleavage[31,33,43,44]. To test the processing of CRISPR-Cas DNA breaks by the DNA end resection machinery, we generated a plasmid-based substrate that can be cleaved by Cas9 or Cas12a at nearly the same position (Fig. 1a). The employed *Streptococcus pyogenes* Cas9 and *Francisella novicida* (*Fn*) and *Moraxella bovoculi* (*Mb*) Cas12a variants linearized the DNA substrate with >90% efficacy (Fig. 1b). As expected, the Cas9 mutants Cas9 D10A (DA, RuvC domain mutant) and Cas9 H840A (HA, HNH domain mutant) formed nicked products with similarly high efficacies (Fig. 1b)[30]. We then monitored DNA end resection in vitro using the *S. cerevisiae* MRE11 complex (Mre11-Rad50, MR) and phosphorylated Sae2 (pSae2) as a co-factor, using a radioactively labeled ssDNA probe that anneals to exposed ssDNA ~92 bp from the generated DSBs if resection has taken place (Fig. 1a)[49]. We note that Xrs2 is dispensable for the DNA end resection activity of the ensemble, and was omitted from the majority of our assays[1,50]. Strikingly, only the Cas12a-treated plasmids were resected, while DNA cut by Cas9 displayed a very low signal (Fig. 1c).

We next investigated whether the Cas9 breaks are resectable by Exo1, i.e., one of the nucleases involved in long-range resection, acting normally downstream of MRE11[5]. While Exo1 completely degraded DNA cut by Cas12a or by the *Eco*RV restriction endonuclease, it could not resect Cas9-dependent breaks, although the DNA was largely linearized by Cas9 (Fig. 1d). These experiments indicate that Cas9-dependent breaks cannot be directly processed by the DNA end resection machinery, in contrast to Cas12a-dependent breaks, which are readily resected (Fig. 1c, d). Cas9 is known to bridge DNA breaks resulting from its endonuclease activity; in vitro, the post-cleavage complex is stable for hours[31,33]. Under our conditions, we have not observed notable processing of Cas9 breaks even after 10 h (Supplementary Fig. 1). Cas12a instead remains only bound to the PAM-proximal end, while the PAM-distal end is released[43,44], possibly explaining why the DNA can be rapidly resected. Our results thus suggest that the DNA end-bridging property of Cas9 might prevent DSB processing.

### A loose DNA end is required for the MRE11 complex activation
To test whether a loose DNA end is required to activate the endonuclease activity of the MRE11 complex, we created plasmid-length DNA substrates of various topological forms, bound with LacI as a non-specific protein block (Fig. 2a–c). The LacI block was placed on circular DNA without an end, on nicked DNA adjacent (nicked circular 2) to a nick or at a distance ~0.9 kbp away (nicked circular 1), or on linear DNA adjacent to an end (Fig. 2b). We next monitored resection downstream of LacI (Fig. 2a). We observed that MR-pSae2 preferentially resected DNA next to a LacI-bound DSB (Fig. 2b, c), despite binding all the substrates with a similar efficacy (Supplementary Fig. 2a). These results suggest that the activation of the endonucleolytic activity of the MRE11 complex requires the presence of loose DNA ends. The DNA termini may nevertheless be shielded by a variety of protein blocks, including physiological (such as the Ku complex, Ku70-Ku80), or non-physiological blocks (streptavidin or LacI), and still be resected with similar efficacies[1,7,8,10,11]. The MRE11 complex does not recognize the chemistry of broken ends, as it incises equally well next to free or loop-capped ends (Fig. 2d–f and Supplementary Fig. 2b). Our results demonstrate that the endonuclease activity of the MRE11 complex is restricted to the vicinity of loose (i.e., unbridged) DNA ends, nevertheless, the ends may be shielded by protein blocks and still be recognized and resected. Such DNA end-sensing capacity prevents spurious cleavage of genomic DNA away from DSB sites by the endonuclease activity of the complex, and may explain the inability of the MRE11 nuclease to process Cas9-dependent breaks, which are known to be bridged[31,33]. In contrast, ends generated by Cas12a, which remains bound only to one end[43,44], are readily resected.

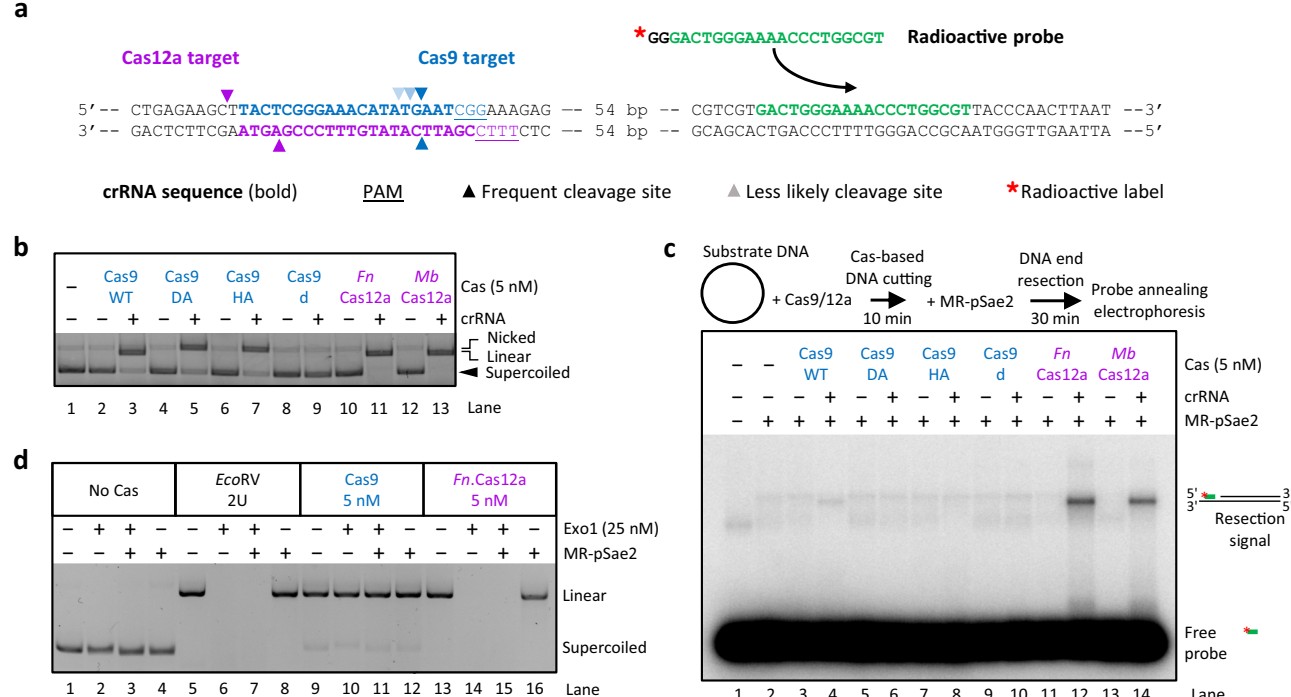

**Fig. 1 | Cas9-dependent breaks are invisible to DNA end resection enzymes. See also Supplementary Fig. 1. a** Sequence of a DNA segment within the ~2.4 kbp circular plasmid-based DNA substrate used for the Cas experiments. The sequence of the crRNAs is shown in bold for both Cas9 and Cas12a. The PAM sequence is underlined, and the arrowheads indicate the putative cleavage sites. The sequence in green corresponds to the radioactive probe that anneals to the substrates if resection has taken place. The red asterisk indicates the position of the radioactive label. **b** Representative agarose gel electrophoresis of the substrates treated with the indicated Cas variants. The separation was performed on a 1% TAE gel in the presence of GelRed. The position of the various products is indicated. DA: Cas9 D10A; HA: Cas9 H840A; d: catalytically inactive Cas9; *Fn*: *Francisella novicida* Cas12a; *Mb*: *Moraxella bovoculi* Cas12a. A representative of two independent

experiments is shown. **c** Top, a schematic overview of the assay. Bottom, representative annealing DNA end resection assay showing resection by the MRE11 complex (25 nM Mre11-Rad50 with 200 nM phosphorylated Sae2, MR-pSae2) of plasmid-based DNA substrate treated with the indicated Cas variants. The probe anneals 92 bp away from the cleavage site for Cas9 and 106 bp for Cas12a. A representative of two independent experiments is shown. **d** Representative Exo1-MR-pSae2-mediated resection of the plasmid-based DNA substrate treated with *Eco*RV, Cas9, or Cas12a, as indicated. DNA was stained with GelRed (Biotium). The DNA products, constituted mostly by mono- and dinucleotides, are not stained effectively by the dye. A representative of two independent experiments is shown. Source data are provided as a Source Data file.

## Bridging of Cas9-dependent DSBs prevents their immediate processing

DSBs formed by ionizing radiation are recognized within seconds by the MRN complex and elicit DNA damage response[51]. Breaks created by Cas9, in contrast, can require up to several minutes in order to be detected[34]. Similarly, ionizing radiation-induced breaks are repaired significantly faster than Cas9-mediated breaks[35,36]. We hypothesized that the observed failure of the MR-pSae2 ensemble to directly resect Cas9-dependent breaks may explain the delay in their processing in vivo. Using two probes annealing on either side of the target site, we found that the Cas9 DSBs were not resected on either side of the break (Fig. 3a, lanes 4, Probe 2 and 9, Probe 1). Strikingly, we observed that *Eco*RV, cutting 20 bp upstream of the Cas9 DNA cleavage complex, restored DNA resection by MR-pSae2 at the monitored site, downstream of Cas9, which is expected to remain attached to the DNA end (Fig. 3a, compare lane 4 with 5 and lane 9 with 10). Similar results were obtained when we inverted the orientation of the guide RNA (gRNA) site, resulting in the opposite orientation of Cas9 on DNA (Supplementary Fig. 3a). Therefore, Cas9 is not a non-specific inhibitor of the employed DNA end resection enzymes, and the inability to resect Cas9 breaks is due to their bridged structure. The end-bridging activity of Cas9 explains the difference in the processing of Cas9 vs. Cas12a breaks (Fig. 3b).

Yeast MRX-pSae2 failed to resect Cas9 breaks similarly as MR-pSae2 (compare Supplementary Fig. 3b with Fig. 3a), in agreement with the dispensable role of Xrs2 in DNA resection[1,50]. We next used

oligonucleotide-based DNA substrates, which were previously observed to be readily incised when bound by various protein blocks near DNA ends[1,7,8,10,11]. The distance of the incision point from the 5′-end was found to be dependent on the length of DNA covered by the protein block (Fig. 3c, d)[6,9]. We also observed efficient incision by MR-pSae2 adjacent to catalytically dead Cas9 (dCas9) bound near the DNA end. The location of the incision site was in agreement with the target sequence of the gRNA, which determines the position of Cas9 on DNA, and the known DNA binding site size of Cas9[52] (Fig. 3c, d). Cas9 was thus sensed as a block by the MRE11 complex, and its nuclease activity was triggered by the nearby ends of the oligonucleotide-based DNA (Fig. 3c, d). The yeast DNA end resection enzymes were used in the above experiments for their high efficacy. However, we note that also the human MRN-pCtIP complex failed to resect Cas9 DSBs on plasmid-length DNA, while its resection activity was restored upon additional DNA cleavage by *Eco*RV (compare Fig. 3e with Fig. 3a). These experiments indicate that the inability to resect Cas9 breaks without further processing generally applies to various DNA end resection enzymes across evolution.

Finally, to generalize our observations, we tested whether Cas9 breaks per se are also refractory to NHEJ. Using purified yeast NHEJ machinery including MRX, Ku70-80, Dnl4-Lif1 and Nej1[53–57], we observed efficient ligation of free DNA ends formed by the *Eco*RV nuclease (Fig. 3f, lane 3 and Fig. 3g). Instead, Cas9 breaks were not a substrate for end-joining (Fig. 3f, lane 6 and Fig. 3g), in agreement with previous experiments performed with the T4 DNA ligase[39]. We note

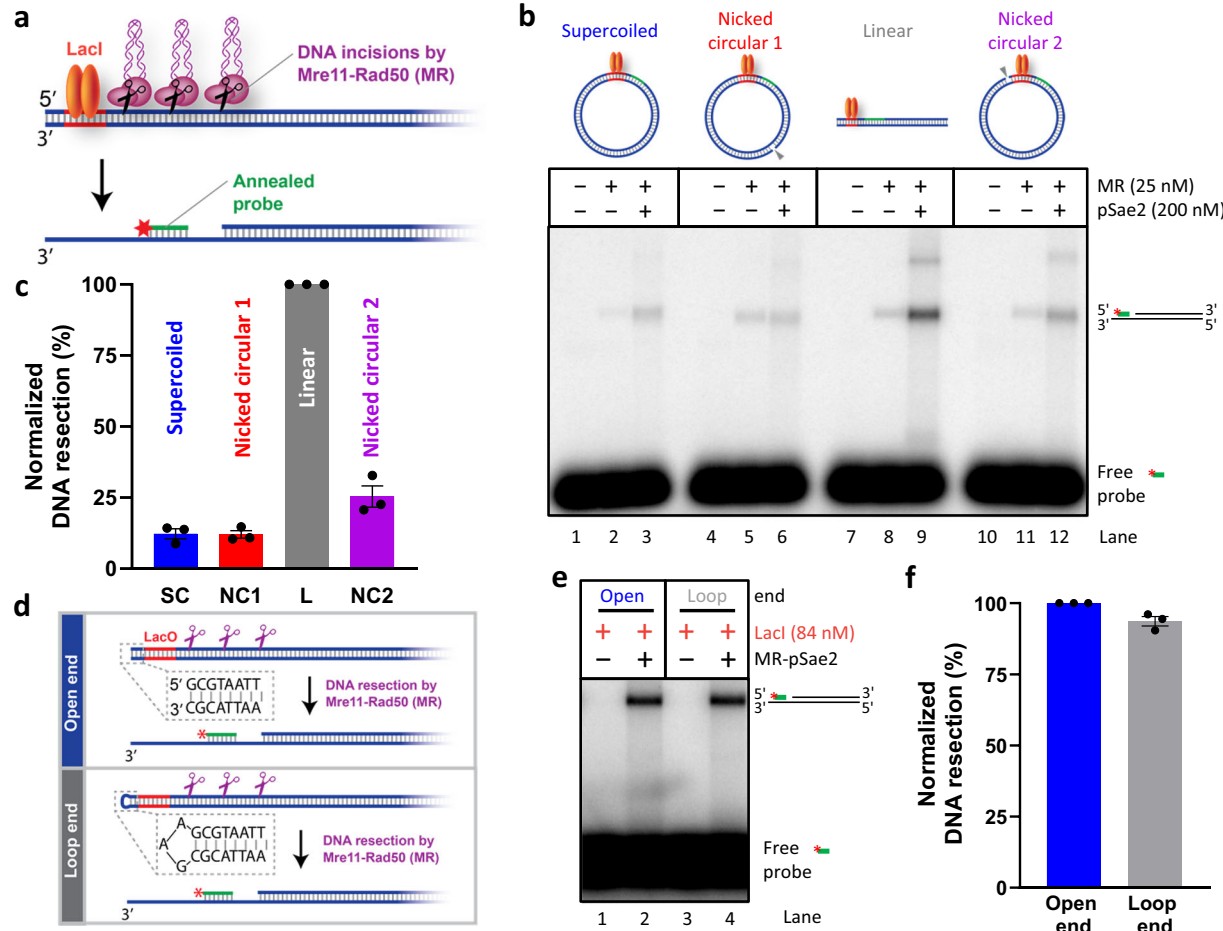

**Fig. 2 | A loose DNA end is required for the MRE11 complex activation.** See also Supplementary Fig. 2. **a** A schematic representation of the assay used to detect Mre11-Rad50 (MR)-mediated resection of a substrate in the presence of dimeric LacI (orange ovals, bound to LacO sequence, in red) acting as a protein block. A radioactively labeled probe (green with a red asterisk indicating the [32]P label) was annealed to the substrate if the 5′-terminated strand was successfully resected. **b** Representative annealing DNA end resection assay on supercoiled, circular nicked, and linear substrates (~2.4 kbp) in the presence of dimeric LacI. In the cartoon, LacI is represented as an orange oval, the probe is represented with a green line and the gray arrowheads represent the position of the nick in the circular nicked substrates (~0.9 kbp away from or adjacent to the LacO for nicked circular 1 and 2, respectively). Phosphorylated Sae2 (pSae2) is a co-factor of the Mre11-Rad50 (MR) complex. We note that all the reactions were incubated with 84 nM LacI (in

monomers) before the addition of the resection factors. **c** Quantitation of experiments such as shown in **b**. $n = 3$; error bars, SEM. Resection efficiency was normalized to the signal of the linear substrate, which was set to 100%. **d** A schematic representation of the assay used to detect MR-mediated resection of open and loop-ended substrates. The red portion of the DNA molecule indicates the position of the LacO site, which binds LacI (not shown). The green line represents the radioactively labeled probe (the red asterisk indicates the [32]P label). **e** Representative annealing assay of MR-mediated resection on open and loop-ended DNA substrates (~2.8 kbp) in the presence of dimeric LacI. **f** Quantitation of experiments such as shown in **e**. $n = 3$; error bars, SEM. Resection efficiency was normalized to the signal of the open-end substrate, which was set to 100%. Source data are provided as a Source Data file.

that the DNA substrate was prepared by PCR, and the DNA ends are therefore not phosphorylated and not ligatable. Ends generated by *Eco*RV are phosphorylated and can be ligated either to each other, or to the ends of the PCR products via a single phosphate group, which can result in larger than substrate DNA ligation products. Our results collectively demonstrate that the HDR and NHEJ DNA end processing machineries cannot readily act on Cas9-mediated DSBs, because they are bridged or otherwise inaccessible (Fig. 3b, g), and implicate the necessity for Cas9 removal prior to repair. These data likely explain the delay in recognition and processing of Cas9 breaks in vivo compared to other sources of DSBs[34].

### HLTF allows resection of Cas9-generated DSBs

Our data indicating that Cas9 breaks are refractory to the MRE11 complex were unexpected, considering that the MRE11 complex is specialized for the removal of protein blocks from DSBs[1,2,7,8,10,11]. Some of the mechanisms capable of Cas9 removal in vitro, as well as from

chromatin in cells, have been identified, including transcription, replication, and the chromatin remodeling complex FACT, showing a notable level of redundancy[39–41]. Since it is well established that Cas9-based genome editing functions also independently of replication and transcription, we searched for additional factors capable of Cas9 removal to facilitate DSB sensing and repair. The human replication fork remodeler HLTF has been recently observed to reduce the efficiency of prime and base editing, together with the expected post-replicative mismatch repair factors[58–60]. Knockdown of HLTF increased the frequency of outcomes that would be expected from a longer residency time of Cas9 on DNA. In particular, HLTF prevented the incorporation of longer DNA segments by reverse transcriptase at the break site, resulting in lower intended editing, as well as a decrease in the unintentional installation of additional sequence modifications from the scaffold sequence[58,59]. The involvement of HLTF was variable and chromatin context-dependent, and it was not apparent which step of the prime editing process was affected. Given the recent

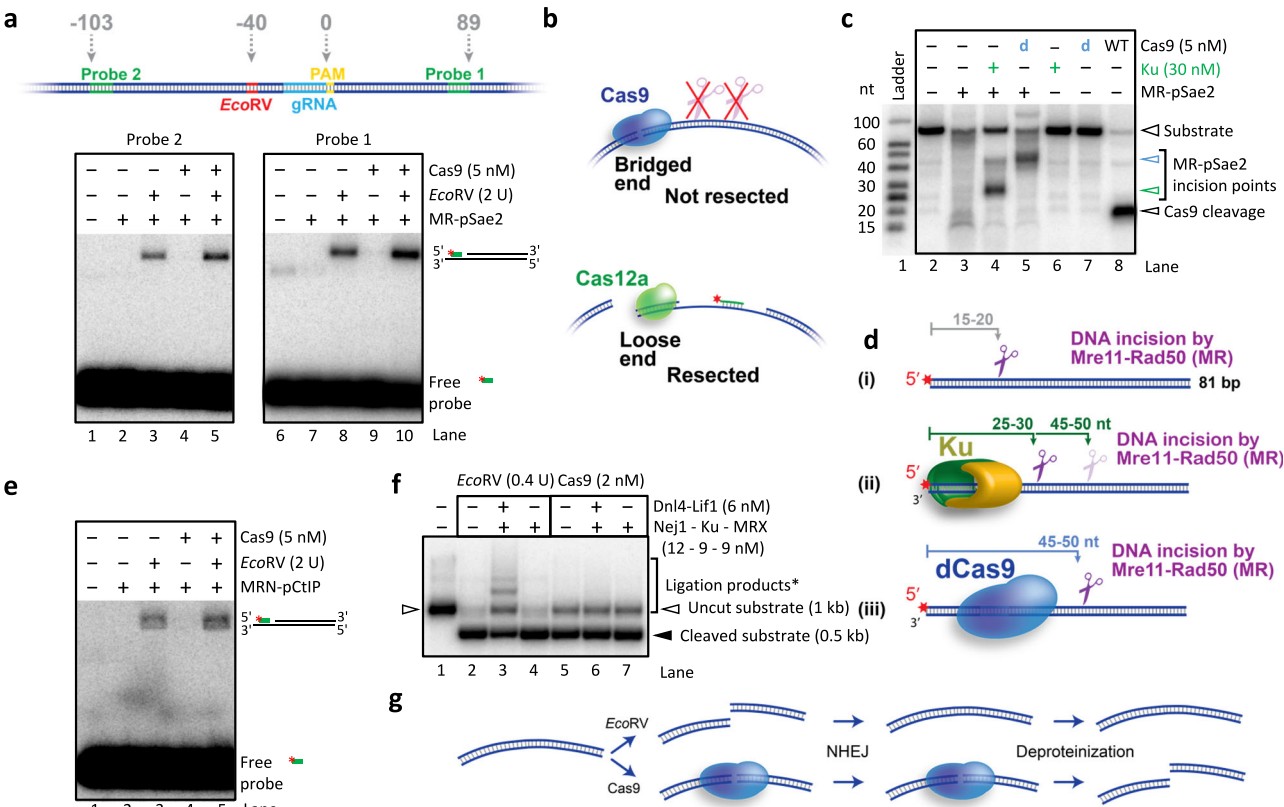

**Fig. 3 | Bridging of Cas9-dependent DSBs prevents their immediate processing. See also Supplementary Fig. 3. a** Comparison of resection on either side of the Cas9 break. Top, a cartoon of the circular plasmid DNA substrate used. The relative distance of the various elements from the boundary between the PAM and the protospacer is indicated in bp. Bottom, representative DNA end resection annealing assays with substrates treated with *Eco*RV and/or Cas9, as indicated. Resection was monitored using a probe on the PAM-distal (left, Probe 2) or PAM-proximal (right, Probe 1) side of the break. A representative of two independent experiments is shown. **b** A model explaining the difference in the processing of Cas9 vs Cas12a breaks by MR and pSae2. Cas9 breaks are not resected because they are bridged. Cas12a breaks are not bridged and, therefore, can be resected. Resection on only one side of the break is depicted for simplicity. **c** Representative MR-pSae2 endonuclease assay with an 81 bp 5′ radioactively labeled oligonucleotide-based substrate in the presence of the Ku complex or catalytically dead Cas9 (dCas9). The

position of the various cleavage products is indicated by arrowheads (refer to **d** for the positions of the DNA incision sites). A representative of two independent experiments is shown. **d** A cartoon of the assay shown in **c**. The positions of the observed DNA incision points are indicated by scissors. The higher transparency of the scissors symbol indicates lower DNA cleavage efficiency. The red asterisk shows the position of the $^{32}$P label. **e** Representative DNA end resection annealing assay using human MRN-pCtIP. A representative of two independent experiments is shown. **f** Representative in vitro non-homologous end-joining (NHEJ) assay of the plasmid-length DNA substrate cleaved with *Eco*RV or Cas9. The bracket (Ligation products*) refers to lanes 2–7, and indicates DNA ligation products, along with a fraction of uncleaved DNA. A representative of two independent experiments is shown. **g** Cartoon of the assay shown in **f**. *Eco*RV-mediated breaks can be ligated in vitro by the yeast NHEJ machinery, while the ends generated by Cas9 are not ligated. Source data are provided as a Source Data file.

identification of HLTF function in the dislocation of nucleotide excision repair complexes[61], independently of DNA replication, we asked whether HLTF could also have the ability to remove Cas9.

To define whether HLTF directly acts on Cas9-DNA post-cleavage complexes, we tested the ability of HLTF to allow resection of the Cas9-dependent DSBs. Additionally, we tested other human motor proteins including translocases implicated in replication fork remodeling, such as SMARCAL1, ZRANB3, FANCJ, FBH1, FANCM and DNA helicases including BLM, WRN, RECQ5, MCM8-9 (together with HROB), and other factors, including selected motor proteins from *S. cerevisiae* (Fig. 4a and Supplementary Fig. 4a). Among the yeast proteins, only Mph1 and Srs2 allowed resection to a notable extent (Supplementary Fig. 4a). The fork remodeling factor HLTF was the only human protein tested capable of extensive Cas9 removal (Fig. 4a), also compared to the tested yeast proteins. This activity did not directly correlate with the translocase activity of these proteins as assayed by Holliday junction branch-migration assay (Supplementary Fig. 4b, c). Human FANCM, the orthologue of yeast Mph1, allowed only limited resection compared to HLTF (Supplementary Fig. 4d). These data demonstrate that among the motor proteins tested, only HLTF has the capacity to efficiently act on Cas9 breaks, as also observed in kinetic experiments

(Supplementary Fig. 4e) and when MR was substituted with MRX (Fig. 4b). Human MRN was also capable to resect Cas9-dependent DSBs exposed by HLTF (Fig. 4b), albeit to a lower extent, which is consistent with the lower activity of the human complex relative to the yeast counterpart in vitro (Supplementary Fig. 4f)[1,2]. Due to the efficiency of Cas9 removal by HLTF and its potential implications for genome editing, we set out to define the underlying mechanism. DNA end resection mediated by HLTF was not observed when dead Cas9 was used or when the resection machinery was omitted from the reaction (Supplementary Fig. 4g), excluding spurious effects of HLTF in our assay. Furthermore, HLTF-mediated stimulation of resection was not dependent on RPA, and RPA per se was not sufficient for the removal of Cas9 (Supplementary Fig. 4h, i). HLTF was also able to allow resection of Cas9-mediated DSBs by Exo1 (Fig. 4c), suggesting that HLTF-mediated removal of Cas9 exposes the DSB ends for processing by multiple pathways (Fig. 4d). We next tested whether the removal of Cas9 by HLTF would make the DNA ends suitable for direct ligation by NHEJ in vitro. A limited end-joining was indeed observed, although notably less efficient than the end-joining of DNA cut by *Eco*RV (Fig. 4e). The reason for the lower efficacy may be repetitive cleavage of DNA by Cas9 in the presence of HLTF, or the known variability in

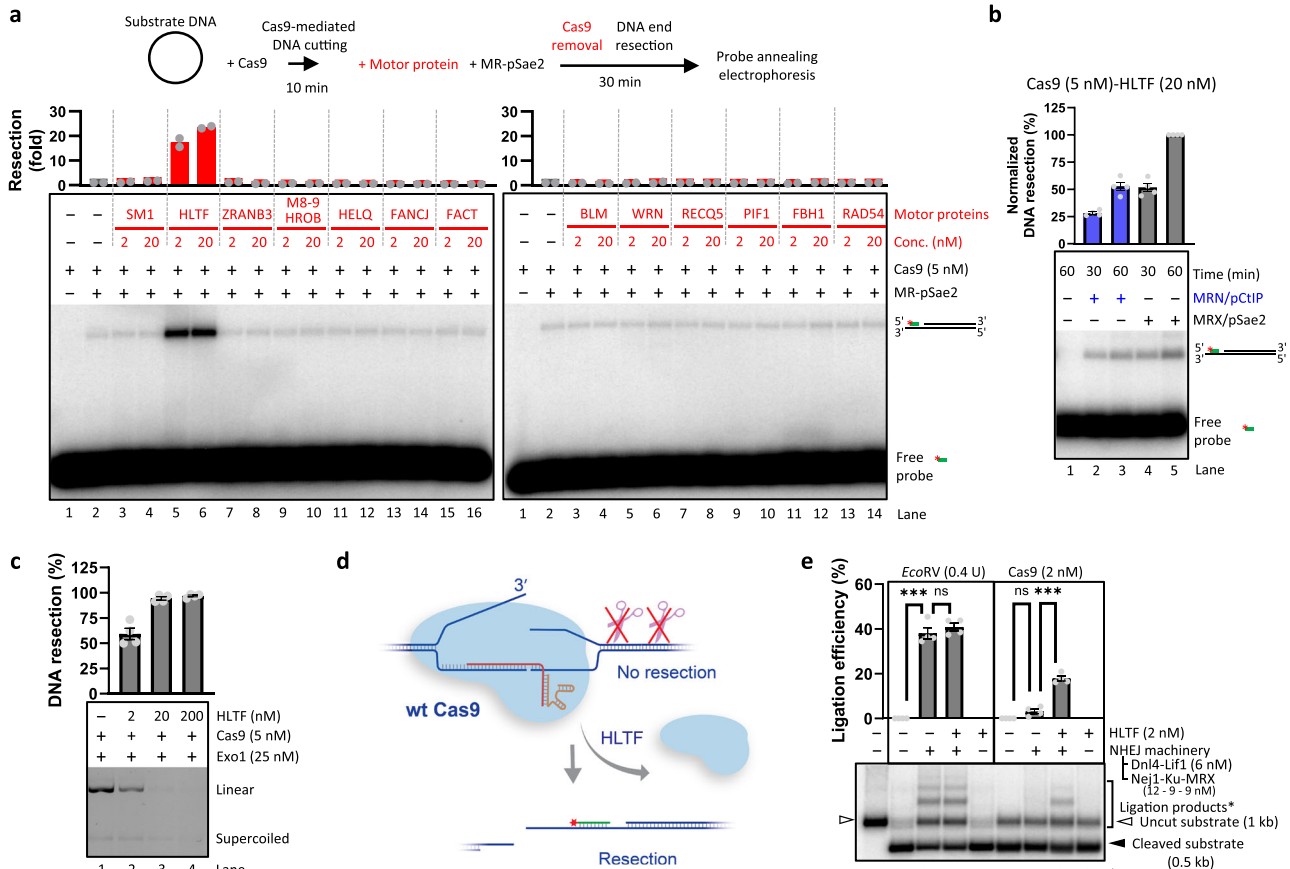

**Fig. 4 | HLTF removes Cas9 from DNA post-cleavage complex. See also Supplementary Fig. 4. a** Annealing DNA end resection assays of Cas9-mediated DNA breaks by MR and pSae2 in the presence of various human DNA translocases. Top, a schematic overview of the assay. Middle: quantitation of resection efficiency expressed as fold increase compared to the "no translocase" sample (lane 2), which was set to 1. Averages shown, $n = 2$. Bottom, representative of two independent experiments. SM1: SMARCAL1; M8-9: phosphorylated MCM8-MCM9; FACT: hSpt16-SSRP1. **b** Annealing DNA end resection assay of Cas9-mediated DNA breaks by yeast MRX-pSae2 or human MRN-pCtIP in the presence of HLTF. Top, quantitation of resection efficiency normalized to the resection obtained by MRX-pSae2 after 1 h, which was set to 100%. $n = 4$; error bars, SEM. Bottom, a representative experiment. **c** Exo1-mediated resection of the plasmid-based DNA substrate cleaved with Cas9 and its dependence on HLTF. Top, quantitation; $n = 4$; error bars, SEM. Bottom, a

representative experiment. DNA was stained with GelRed (Biotium). The DNA products, constituted mostly by mono- and dinucleotides, are not stained effectively by the dye. **d** A cartoon depicting HLTF-mediated Cas9-removal. HLTF is capable of removing Cas9 from the DNA post-cleavage complex, thus allowing DNA end resection of the DSB. **e** In vitro non-homologous end-joining of *Eco*RV or Cas9-dependent DNA breaks, and the effect of HLTF. Top, quantitation of ligation efficiency. $n = 4$; error bars, SEM. Bottom, a representative experiment. The bracket (Ligation products*) refers to lanes 2–7, and indicates DNA ligation products, along with a proportion of uncleaved DNA. (ns, non significant) $p = 0.3221$ (lanes 3 and 4) and $p = 0.0520$ (lanes 6 and 7), (***) $p = 0.0006$ (lanes 2 and 3) and $p = 0.0007$ (lanes 7 and 8), two-tailed paired $t$ test. The amount of uncleaved substrate in lanes 2 and 6 was used as background. Source data are provided as a Source Data file.

Cas9 incision points of the non-target strand leading to incompatible DNA ends[30]. Together, our data demonstrate that Cas9 can be removed from the post-cleavage complex by the motor activity of HLTF, which allows subsequent DSB processing and repair, such as DNA end resection, leading to HDR, as well as NHEJ.

## HLTF dislocates Cas9-DNA post-cleavage complexes and allows Cas9 to act catalytically

The stability of Cas9-DNA post-cleavage complexes explains why Cas9 does not function repetitively in vitro[31]. An excess of Cas9 over the DNA substrate is required for the majority of the DNA molecules to be cleaved[32,33]. To test whether HLTF can dislocate Cas9 from DNA post-cleavage complexes, we performed a multi-turnover assay, in which a limited concentration of Cas9 was used to cleave an excess of plasmid DNA. In case HLTF activity resulted in incomplete dissociation of Cas9 from the DSBs, such as when Cas9 would remain bound to one side of the break, HLTF would have no effect on Cas9 cleaving the excess substrate (scenario 1 in Fig. 5a). In contrast, if HLTF completely dissociates Cas9 from both ends, a single Cas9 polypeptide would in

principle be able to act sequentially on multiple DNA substrate molecules in a repetitive fashion (scenario 2 in Fig. 5a). Consistently with the known single-turnover properties of Cas9, limited concentration of Cas9 (0.5 nM) resulted in only ~20% cleavage of the excess plasmid DNA substrate (1 nM, in molecules, Fig. 5b, lane 2 in Supplementary Fig. 5a). HLTF strongly promoted the reaction, leading to nearly 100% linearization of the substrate with the substoichiometric concentration of Cas9 (Fig. 5b and Supplementary Fig. 5a). The experiments suggested that HLTF completely dissociates Cas9 from the break, and allows it to act repetitively, confirming scenario 2 in Fig. 5a. Lowering the concentration of Cas9, while keeping the DNA substrate concentration constant, in the multi-turnover assay resulted in one molecule of Cas9 being able to cleave up to ~18 DNA molecules in the presence of HLTF (Fig. 5c and Supplementary Fig. 5b). For Cas9 to act repetitively, ATP-hydrolysis was required (Fig. 5b and Supplementary Fig. 5a), suggesting that the motor activity of HLTF is needed to dislocate Cas9, in a manner that is reminiscent of its role in the eviction of NER complexes and in fork reversal[61–63]. Together, our data suggest that HLTF completely dissociates Cas9 from DNA, resulting in a clean

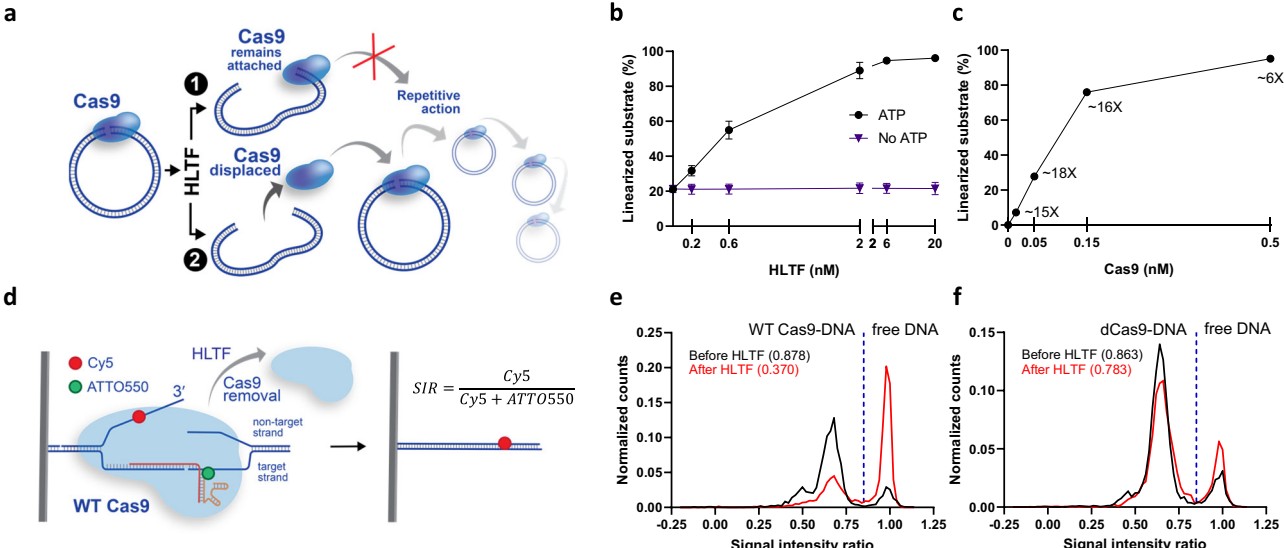

**Fig. 5 | HLTF removes Cas9 allowing it to cut multiple targets. See also Supplementary Fig. 5. a** A schematic of the multi-turnover assay used to determine the mechanism of Cas9-removal by HLTF. **b** Quantitation of multi-turnover experiments of Cas9 and DNA (1 nM, in molecules) in the presence of increasing concentrations of HLTF as shown in Supplementary Fig. 5a. $n = 3$; error bars, SEM. **c** Quantitation of multi-turnover experiments with DNA (1 nM, in molecules), varying concentration of Cas9 (as indicated), and HLTF (20 nM) as shown in Supplementary Fig. 5b. $n = 4$; error bars, SEM. The values next to each data point represent the fold stimulation of Cas9 cleavage by HLTF compared to the conditions without HLTF (lane 2 in Supplementary Fig. 5b; 16 ± 1% linearization; $n = 4$; SEM) at the given concentration. We note that the fold stimulation for the 0.5 nM sample (lane 6, Supplementary Fig. 5b) is limited by the amount of substrate used. **d** A cartoon of the single-molecule fluorescence-based Cas9 dissociation assay. The DNA is visualized by Cy5-labeling, while Cas9 is visualized via ATTO550 labeling of the tracrRNA. A signal intensity ratio value of 0.85 was used as a threshold between Cas9-bound DNA (<0.85) and unbound DNA (>0.85). The formula used for the calculation of the signal intensity ratio (SIR) is presented on the right. **e**, **f** Single-molecule fluorescence-based assay monitoring the removal of wild type (**e**) or catalytically dead (**f**) Cas9 by HLTF. The distribution of the normalized counts of more than 1000 molecules from two independent experiments is presented. The numbers in parentheses indicate the fraction of bound molecules. Source data are provided as a Source Data file.

DSB and allowing the freed Cas9 enzyme to act on other DNA molecules.

To directly demonstrate that HLTF removes Cas9 from DNA, we employed a single-molecule fluorescence-based assay, in which a Cy5-labeled DNA molecule was bound to a surface via biotin-streptavidin and targeted with a Cas9-gRNA complex labeled with ATTO550 (Fig. 5d and Supplementary Fig. 5c)[30]. When HLTF was added to the DNA substrate previously reacted with wild type Cas9, the signal corresponding to free DNA intensified, indicative of Cas9 removal from the DNA by HLTF (Fig. 5e). Similar results were obtained when using a substrate labeled with Alexa750 on the other side of the break (Supplementary Fig. 5d, e). HLTF was able to remove wild type Cas9 over time (Supplementary Fig. 5f). Unexpectedly, efficient removal was not observed when nucleolytically inactive Cas9 was used (Fig. 5f), suggesting that HLTF preferentially dislocates Cas9-DNA post-cleavage complexes, in contrast to previously described processes mediating Cas9 removal[39–41].

### HLTF acts on the 3'-end released by the RuvC domain of Cas9

We next analyzed the capacity of HLTF to remove the point mutant nickase variants of Cas9. The H840A mutation disrupts the HNH nuclease domain of Cas9, which is unable to cleave the target DNA strand[30]. The D10A mutations in the RuvC-like nuclease domain of Cas9 instead prevent cleavage of the non-target DNA strand (Fig. 6a, b). The single-molecule fluorescence experiments revealed that HLTF was able to remove the Cas9 H840A nickase (Fig. 6a), while it was ineffective in the removal of Cas9 D10A (Fig. 6b). Annealing experiments and the crystal structure of the Cas9 DNA post-cleavage complex revealed that the cleaved target strand, duplexed with gRNA/crRNA, remains buried in the Cas9 structure, while the 3'-end of the non-target strand is exposed on the surface of the protein[31,64,65]. As the Cas9 H840A but not the Cas9 D10A nickase releases the 3' strands (Fig. 6a, b, cartoon at the top of the panels), the results suggested that

the displaced 3'-end may facilitate Cas9 removal by HLTF. The data are also consistent with HLTF being poorly capable of removing catalytically inactive Cas9 (Fig. 5f). Similarly, in the multi-turnover assay, the addition of HLTF was able to induce multi-turnover behavior with Cas9 H840A but not with Cas9 D10A (Fig. 6c, d). Consistently, Exo1 can extend the nick generated by Cas9 HA after it was exposed to HLTF (Supplementary Fig. 6). The requirement for the free 3'-end differentiates HLTF-mediated Cas9 removal from previously observed processes including transcription, FACT-mediated nucleosome rearrangement and replication that are also able to remove catalytically dead Cas9[39–41].

### HLTF knockdown induces Cas9 H840A-mediated activation of DNA damage response

In cells, the two Cas9 nickase variants display distinct behaviors. Cas9 H840A fails to elicit a strong DNA damage response, while the D10A mutant induces checkpoint signaling as proficiently as wild type Cas9[66]. Cas9 H840A is also less efficient at triggering homologous recombination[45,66]. If HLTF removes Cas9 H840A, the nick may be rapidly sealed by the cellular single-stranded DNA break repair machinery, masking the effects of Cas9 H840A. To test whether the ability of HLTF to remove Cas9 H840A, but not D10A, could account for this difference, we generated MCF10A-derived cell pools harboring a doxycycline-inducible Cas9 H840A or D10A constructs. We used a gRNA complementary to the Alu and LINE1 repetitive elements with thousands of target sites across the human genome[67]. We then monitored the DNA damage response in cells treated or not with siHLTF (Fig. 6e). We observed that Cas9 D10A activated phosphorylation of H2AX as well as RPA at S4/S8, two markers of the DNA damage response. The checkpoint signaling was unaffected by siHLTF (Fig. 6e, lanes 6 and 7). In contrast, Cas9 H840A cells did not display significant DNA damage response activation (Fig. 6e, lane 2), in agreement with previous data[66]. Notably, the knockdown of HLTF in Cas9 H840A cells

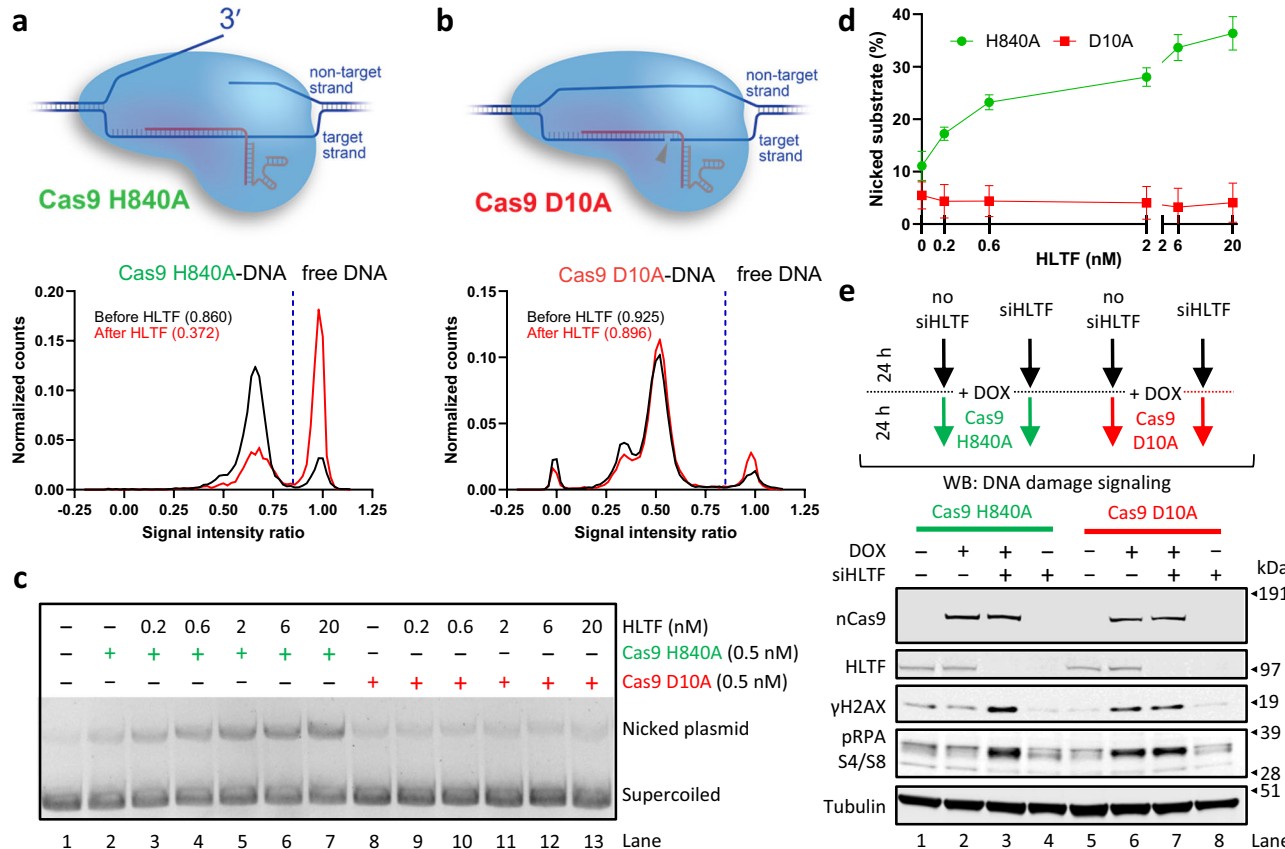

**Fig. 6 | HLTF acts on 3′-ends released by Cas9 after DNA cleavage.** See also Supplementary Fig. 6. **a** Single-molecule fluorescence-based assay showing the removal of Cas9 H840A by HLTF. Top, a cartoon of the Cas9 H840A-DNA post-cleavage complex. Cas9 H840A fails to cleave the target strand but is proficient in creating the 3′ ssDNA upon cleavage of the non-target strand. Bottom, the distribution of the normalized counts of >1000 molecules from two independent experiments is presented. The dashed blue line indicates the separation between DNA only (>0.85) and Cas9-bound DNA (<0.85). The numbers in parentheses indicate the fraction of bound molecules. **b** Single-molecule fluorescence-based assay showing the removal of Cas9 D10A by HLTF. Top, a cartoon of the Cas9 D10A-DNA post-cleavage complex. Cas9 D10A only cleaves the target strand. Bottom, the distribution of the normalized counts of more than 1000 molecules from two

independent experiments is presented. The dashed blue line indicates the separation between DNA only (>0.85) and Cas9-bound DNA (<0.85). The numbers in parentheses indicate the fraction of bound molecules. **c** Representative multi-turnover assay with Cas9 H840A and D10A nickase variants in the presence of increasing concentration of HLTF. **d** Quantitation of experiments such as shown in c. *n* = 4; error bars, SEM. **e** DNA damage response after induction of breaks using a guide targeting the Alu and LINE1 repetitive elements. Top, schematic overview of the cellular assay. Bottom, representative western blots from two independent experiments. Cells were treated 48 h before collection with siRNA against HLTF, as indicated. Expression of the indicated Cas9 nickase was induced with doxycycline (DOX) 24 h before collection, as indicated. pRPA S4/S8: antibody against phosphorylated Ser4 and Ser8 of RPA32. Source data are provided as a Source Data file.

restored checkpoint signaling to the same levels as observed in cells expressing Cas9 D10A (Fig. 6e, lane 3). HLTF-mediated removal of Cas9 H840A thus explains the different cellular impacts of the two Cas9 nickase variants.

**The HIRAN domain of HLTF is important for Cas9 removal**
The N-terminal part of HLTF contains the HIRAN domain that was shown to bind a 3′-OH moiety at the end of ssDNA, with additional contacts along the DNA backbone[63,68] (Fig. 7a). The binding of the HIRAN domain was proposed to direct HLTF to the 3′ terminated nascent DNA strand to initiate replication fork reversal, or the 3′-end of the incised DNA in HLTF-stimulated nucleotide excision repair[61,68]. We established that Cas9 removal by HLTF requires the release of the 3′-terminated ssDNA from the Cas9-DNA cleavage complex (Fig. 6). We thus reasoned that the HIRAN domain could be responsible for the recognition of the 3′ terminus, and HLTF could translocate on this strand in the 3′−5′ direction, leading to the dislocation of Cas9. To test this hypothesis, we purified the HLTF N90A-N91A mutant (HLTF NANA) that disrupts the function of the HIRAN domain (Fig. 7a and Supplementary Fig. 7a)[61,63]. The HLTF NANA mutant exhibited a similar DNA branch-migration activity on a Holliday junction substrate

compared to the wild type protein (Fig. 7b, Supplementary Fig. 7b, c), suggesting that its motor activity is not compromised by the NANA mutations. However, the HLTF NANA mutant largely failed to remove Cas9 from DNA upon cleavage, as observed in the single-molecule fluorescence experiments (Fig. 7c). This defect was also very apparent in the multi-turnover assay (Fig. 7d, e, compare with Fig. 5b and Supplementary Fig. 5a). The addition of HLTF NANA had only a minor effect on the stimulation of the apparent activity of Cas9, while wild type HLTF induced multi-turnover behavior of Cas9 at much lower concentrations. Consequently, the HLTF NANA mutant was also notably impaired in its capacity to facilitate resection of Cas9-dependent DNA breaks by MR-pSae2 (Fig. 7f).

To further support the involvement of the HIRAN domain of HLTF in the removal of Cas9, we purified the first 180 amino acids of the HLTF protein, corresponding to the HIRAN domain, either in the wild type (HIRAN WT), or mutated (HIRAN NANA) form (Fig. 7g and Supplementary Fig. 7d)[69]. We observed that in the multi-turnover assay, the wild type HIRAN domain inhibited the removal of Cas9 by HLTF, while the HIRAN NANA variant inhibited the reaction much less effectively (Fig. 7h, i). Correspondingly, HIRAN WT acted in a dominant negative manner and inhibited DNA end resection of Cas9-dependent

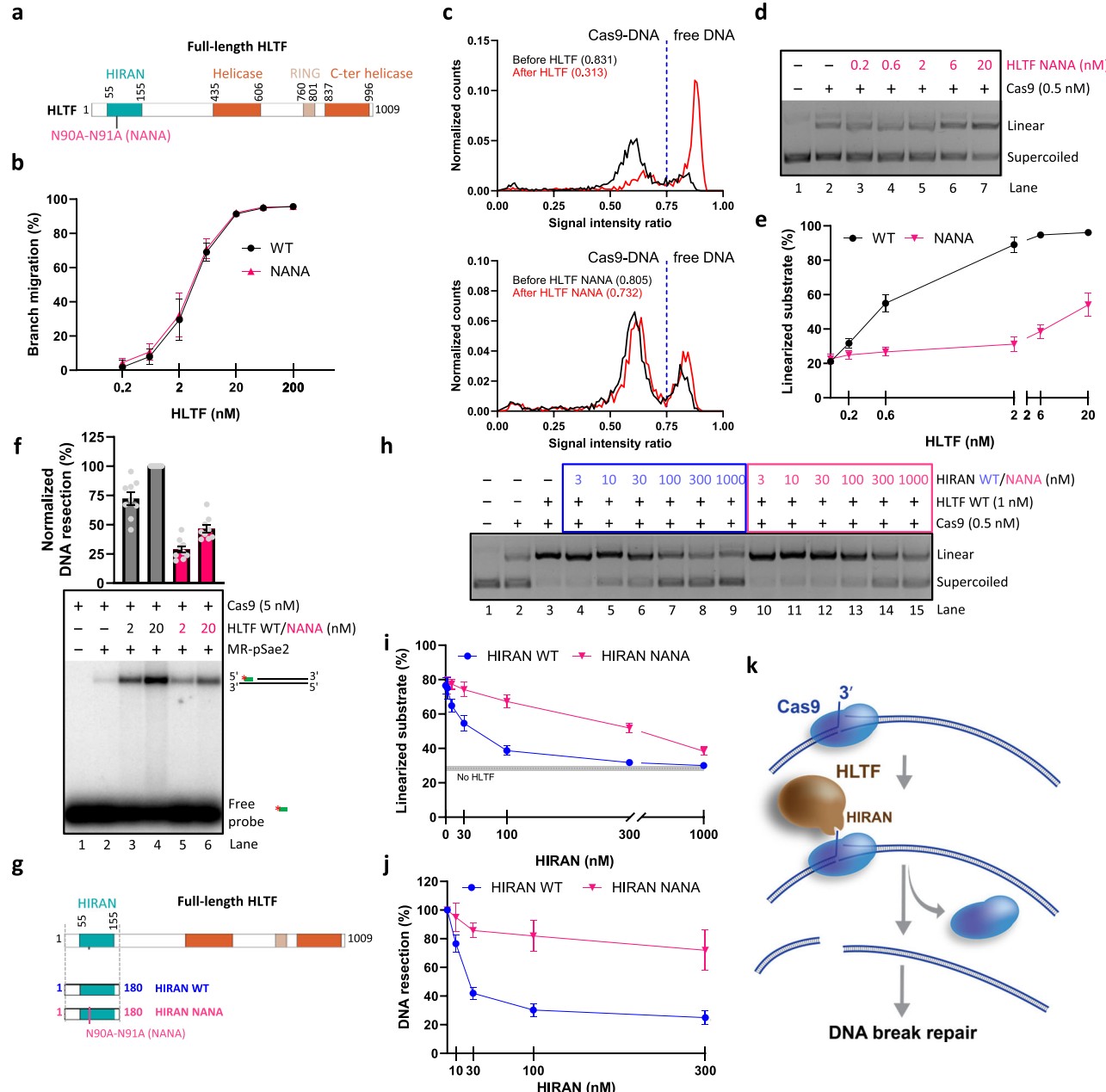

**Fig. 7 | The HIRAN domain of HLTF is important for Cas9 removal. See also Supplementary Fig. 7. a** A cartoon representation of the primary structure of HLTF. The position of the HIRAN domain mutations (N90A-N91A, NANA) is indicated. **b** Quantitation of DNA branch-migration experiments such as shown in Supplementary Fig. 7b, c. n = 4 (WT) and n = 3 (NANA); error bars, SEM. **c** Single-molecule fluorescence-based assays showing the removal of wild type Cas9 by HLTF WT (top) and NANA mutant (bottom). The distribution of the normalized counts of more than 1000 molecules from two independent experiments is presented. The dashed blue line indicates the separation between DNA only (>0.75) and Cas9-bound DNA (<0.75). **d** Representative Cas9 multi-turnover assay in the presence of Cas9, 1 nM DNA, and increasing concentration of HLTF NANA mutant. **e** Quantitation of experiments such as shown in **d**. n = 3; error bars, SEM. The HLTF WT data is reproduced from Fig. 5b (ATP). **f** Annealing DNA end resection assay of Cas9 breaks by MR-pSae2 in the presence of HLTF WT or NANA mutant. Top, quantitation. n = 9; error bar, SEM. Bottom, a representative experiment. DNA resection efficiency was normalized to the sample with 20 nM HLTF WT, which was set to 100%. **g** A cartoon representation of the domain structure of the HLTF protein and the purified HIRAN domain variants (1–180). The position of the NANA HIRAN domain mutations is indicated. **h** Representative Cas9 multi-turnover assay in the presence of Cas9, 1 nM DNA, HLTF, and increasing concentration of WT and NANA mutant HIRAN domain. **i** Quantitation of experiments such as shown in **h**. n = 5; error bars, SEM. The gray line indicates the fraction of linearized substrate in the absence of HLTF. **j** Quantitation of annealing DNA end resection assays such as shown in Supplementary Fig. 7e. n = 4 (WT) and n = 3 (NANA); error bars, SEM. **k** A model for HLTF-mediated Cas9 removal from a DNA post-cleavage complex. HLTF recognizes the 3′-end released by Cas9 after DNA cleavage through its HIRAN domain. The motor activity is subsequently engaged and leads to Cas9 displacement allowing DNA repair. Source data are provided as a Source Data file.

breaks in the presence of HLTF, while HIRAN NANA had a much lesser effect (Fig. 7j and Supplementary Fig. 7e). Consistently when testing the binding of the HIRAN WT or NANA domains to Cas9 HA or DA DNA post-cleavage complexes (Supplementary Fig. 7f, g), we observed the

best binding using the HIRAN WT domain on top of Cas9 HA, which generates the 3′ overhang.

These data collectively support a model in which HLTF acts on the Cas9 cleavage complex through its HIRAN domain, which binds the

exposed 3′ strand of the Cas9-DNA post-cleavage complex (Fig. 7k). HLTF then engages its motor activity in order to remove Cas9, which enables downstream processing and repair (Fig. 7k).

## Discussion

The MRE11 complex is specialized for the removal of protein blocks from DNA ends[6]. Such capacity makes it a prime candidate for the removal of Cas9 from DNA breaks. Indeed, Cas9-dependent breaks were mapped genome-wide by the localization of MRE11 adjacent to the breaks[42]. We report here that, unexpectedly, neither the yeast (MRX) nor the human (MRN) complex is capable of sensing and initiating DNA end resection of Cas9-dependent DSBs without additional prior processing. The inability of MRN/X to act on Cas9 breaks is explained by the requirement for a loose end for the MRE11 complex activation. While Cas9 is known to bridge DNA ends[32,33], Cas12a, in contrast, remains bound to the PAM-proximal DSB, and the other end of the DNA is released[43,44]. In accordance with our model, Cas12a breaks are readily resectable by the MRE11 complex, while Cas9 breaks are refractory. Our data suggest that Cas9 breaks require further processing before they can be addressed by the DNA end resection machinery including MRX/N.

Mechanisms involved in Cas9 removal have been identified, including transcription, replication, and chromatin remodeling[39–41]. Cas9 residency time at the target site after break formation has been shown to affect the outcome of the repair[37,38]. Recent screens for factors that influence prime editing uncovered the role of HLTF in suppressing the desired outcome and, more generally, events that result from longer residency time of the Cas9-reverse transcriptase fusion on DNA, although the mechanism of HLTF involvement was unclear[58,59]. Using ensemble and single-molecule data, we reveal here that HLTF directly removes Cas9 from DNA upon cleavage, and enables it to act sequentially on multiple DNA molecules. We show that HLTF-dependent Cas9 removal from the DNA post-cleavage complex requires its HIRAN domain to bind 3′-terminated ssDNA strands[63,68]. We propose that HLTF then employs its 3′−5′ translocase activity to move along DNA, resulting in the complete dislocation of Cas9. Our data suggest that HLTF can remove Cas9 in a manner that is reminiscent of its role in fork reversal[63,68] and nucleotide excision repair[61].

The uncovered mechanism of Cas9 removal by HLTF fundamentally differs from the previously described processes due to the requirement for the displaced 3′-end from the Cas9 post-cleavage complex (Fig. 6)[39–41]. Consequently, HLTF can act on the Cas9 H840A but not on the Cas9 D10A nickase, because only the H840A variant releases the 3′-end after DNA cleavage[31]. Previous studies noted different impacts of the two Cas9 nickase variants on the activation of homologous recombination and the DNA damage response in human cells[46,70]. While Cas9 D10A triggered robust checkpoint signaling and recombination, the Cas9 H840A variant triggered only a minimal response[66]. We show that HLTF-catalyzed removal of Cas9 H840A masks its impact on checkpoint signaling. In the absence of HLTF, both Cas9 variants elicit a similar response. We hypothesize that once HLTF dislocates Cas9 H840A, the uncovered nick is rapidly sealed by the cellular ssDNA break repair proteins. In contrast, nicks that persist in DNA until the arrival of DNA replication forks, such as those induced by Cas9 D10A, cause replication-associated DNA breaks, which are responsible for the majority of the DNA damage response. Consistently, Cas9 H840A nickase triggers replication-associated DSBs only in the absence of HLTF.

## Methods

### Generation of expression vectors and purification of recombinant proteins

**Cas9 and Cas12a purification.** The *Streptococcus pyogenes* (*Sp*) Cas9 expression vector, pMJ922[71] (Addgene #78312) encodes for Cas9 tagged with hexahistidine and maltose binding protein tags at the N-terminal (6xhis-MBP) and with HA, GFP and three nuclear localization signals at the C-terminal (HA-2xNLS-GFP-NLS). A TEV protease cleavage site is present between the 6-his-MBP tag and Cas9, allowing the removal of the N-terminal tag. Point mutations for nickases (Cas9_D10A, Cas9_H840A) and catalytically dead Cas9 (Cas9_D10A, H840A) were introduced by inverse PCR and confirmed by DNA sequencing. The *Francisella novicida* (*Fn*) Cas12a was expressed using pDS015[72]. *Moraxella bovoculi* (*Mb*) (WP_052585281) Cas12a gene was sourced as synthetic genes, codon optimized for *Escherichia coli* (*E. coli*) (GeneArt, Thermo Fisher Scientific), and cloned into the pET-1B expression vector (Addgene plasmid #29653) by LIC generating the pMS026 expression construct (6xHis-TEV-*Mb*Cas12a). Cas9 and Cas12a were purified using NiNTA affinity chromatography, a heparin purification step, and size exclusion chormatography[30,52,73,74]. In brief, Cas9 and Cas12a constructs were expressed in *E. coli* BL21 Rosetta2 (DE3) cells (Novagen). Cells were lysed by ultrasonication in lysis buffer containing 20 mM Tris pH 8, 500 mM NaCl, 5 mM imidazole, 1 μg/ml Pepstatin and 200 μg/ml AEBSF. Clarified lysate was applied to a 10 ml NiNTA (Sigma-Aldrich) affinity column. The column was washed with 20 mM Tris pH 8.0, 500 mM NaCl, 10 mM imidazole, and bound protein was eluted with an imidazole gradient to 250 mM. Eluted protein was dialyzed against 20 mM HEPES pH 7.5, 250 mM KCl, 10% glycerol, 1 mM dithiothreitol, 1 mM EDTA (Cas9), or 20 mM HEPES pH 7.5, 250 mM KCl, 1 mM dithiothreitol, 1 mM EDTA (Cas12a) overnight at 4 °C in the presence of TEV protease to remove the 6xhis-MBP (Cas9) or 6xhis (Cas12a) affinity tags. The cleaved protein was loaded on a HiTrap HP Heparin column (Cytiva) and eluted with a linear salt gradient to 1 M KCl. The elution fractions containing the protein were pooled, concentrated, and loaded on a size exclusion Superdex 200 (16/600) column (Cytiva) in 20 mM HEPES-KOH pH 7.5, 500 mM KCl, 1 mM dithiothreitol yielding pure, monodisperse proteins. The protein was aliquoted, flash-frozen in liquid nitrogen, and stored at −80 °C.

**Yeast MRX and phosphorylated Sae2 purification.** Unless indicated otherwise, all proteins purified from *Spodoptera frugiperda* cells (*Sf*9, repository of the Institute of Molecular Cancer Research, Zurich, Switzerland) were expressed for 52 h using recombinant baculoviruses obtained from the indicated vectors using the Bac-to-bac system (Invitrogen) following manufacturer's instructions. The cells were collected, washed with PBS 1× (Gibco), frozen in liquid nitrogen, and stored at −80 °C until the day of the purification. The *Saccharomyces cerevisiae* Mre11-Rad50-Xrs2 (MRX) complex was expressed in in *Spodoptera frugiperda* 9 (*Sf*9) cells using pTP391[75] (a kind gift of Tanya Paull, University of Texas at Austin), expressing his-tagged Mre11, pFB-Rad50, expressing codon-optimized untagged Rad50[76] and pTP694[75] (Tanya Paull, University of Texas at Austin) expressing Xrs2-FLAG and purified by sequential NiNTA (Qiagen) and FLAG (Sigma) affinity purification[77]. The Mre11-Rad50 (MR) complex was expressed as the MRX complex with the omission of the baculovirus expressing Xrs2, and purified by NiNTA affinity chromatography followed by ion exchange chromatography (HiTrap SP HP and HiTrap Q HP columns, both Cytiva) on an AKTA system[50]. Phosphorylated Sae2 (pSae2) was expressed in *Sf*9 cells using the pFB-MBP-Sae2-his construct and purified by amylose and NiNTA affinity chromatography with phosphatase inhibitors[1,78]. The MBP-tag was removed by digestion with PreScission Protease.

**Human MRN and phosphorylated CtIP purification.** The human MRN complex was purified from *Sf*9 cells using the pTP17[79], pFB-RAD50-FLAG[2], and pTP36[79] vectors coding for MRE11-6xhis, RAD50-FLAG, and NBS1, respectively (pTP17 and pTP36 were provided by Tanya Paull, University of Texas at Austin, Austin, TX), and purified by NiNTA and FLAG affinity purification[2]. Phosphorylated CtIP (pCtIP) was purified by amylose and NiNTA purification from cells infected with baculovirus

produced from the pFB-MBP-CtIP-his vector[2]. The MBP tag was removed by PreScission Protease before the NiNTA purification step.

**LacI purification.** The construct for the expression of his-tagged dimeric LacI, pMALT-P_LacI(1-340)_6xhis, was produced by cloning into a pMALT-P vector (Kowalczykowski laboratory, UC Davis) the coding region for LacI (amino acids: 1 to 340) obtained by PCR from the pMALT-P vector with the following primers: LacI-for and LacI_6xhis-rev (see Supplementary Table 1 for primer sequences). The LacI_6xhis-rev primer added a sequence coding for in-frame 6xhis tag to the C-terminus of the protein. The construct was transformed in BL21 (DE3) pLysS cells and the culture was grown overnight at 37 °C. Expression was induced at an OD of 1.5 with 1 mM isopropyl β-D-1-thiogalactopyranoside (IPTG) for 3 h at 37 °C. Cells were harvested by centrifugation at 3000 × g for 15 min, washed with cold PBS, and centrifuged again at 3000 × g for 15 min. The pellet was snap-frozen in liquid nitrogen and stored at −80 °C until processing. All subsequent steps were performed at 4 °C. For purification, the pellet was resuspended in NiNTA Wash Buffer containing 50 mM Tris-HCl pH 7.5, 1 mM dithiothreitol, 10% glycerol, 500 mM NaCl and 1 mM phenylmethylsulfonyl fluoride (PMSF), supplemented with protease inhibitor cocktail (P8340, Sigma-Aldrich, 1:300). The mixture was sonicated, and the soluble extract was cleared by centrifugation for 30 min at 50,000 × g at 4 °C and filtered with 0.44 μM filters. The clarified soluble extract was loaded on a pre-equilibrated 1 ml HisTrap HP column (Cytiva) and washed with NiNTA Wash Buffer and later with NiNTA Wash Buffer supplemented with 20 mM imidazole. The protein was eluted in 250 μl fractions with a 5 ml gradient, from 20 to 300 mM imidazole. The fractions containing the protein were pooled and desalted using a HiTrap Desalting column (Cytiva) equilibrated in Buffer A (50 mM Tris-HCl pH 7.5, 1 mM dithiothreitol, 10% glycerol, 50 mM NaCl and 1 mM PMSF). The desalted protein was loaded on a HiTrap Heparin column (Cytiva) and washed extensively with Buffer A to remove the IPTG bound to the protein. Dimeric LacI was eluted with a 10 ml salt gradient in buffer A (from 50 to 1 M NaCl), collecting 250 μl fractions. The fractions containing the protein were pooled, aliquoted, and snap-frozen. The concentration in monomers was calculated by measuring the absorbance at 280 nm using a predicted molar extinction coefficient of 23045 M$^{-1}$ cm$^{-1}$.

**Ku complex (Ku70-Ku80) purification.** The *Saccharomyces cerevisiae* Ku heterodimer (Ku70-Ku80) was expressed in *Sf*9 insect cells infected using pFB-MBP-Ku70-his and pFB-Ku80-FLAG coding for MBP- and his-tagged Ku70 and FLAG-tagged Ku80, respectively. The complex was purified using amylose and FLAG affinity resins[7]. The MBP tag was removed before the FLAG purification step using PreScission Protease.

**Yeast Dnl4-Lif1 heterodimer purification.** The sequences coding for Dnl4 and Lif1 were amplified by PCR from *Saccharomyces cerevisiae* genomic DNA (yWH436 strain[80]) using FlagDNL4FO with FlagDNL4RE and LIF1FO with LIF1RE, respectively. The products were digested using *Bam*HI-*Xma*I (Dnl4) and *Nhe*I-*Xho*I (Lif1), and inserted into pFB-MBP-Sgs1 digested with same enzymes to generate pFB-MBP-Lif1 and pFB-FLAG-Dnl4, respectively. Dnl4 and Lif1 were coexpressed, and the complex was purified with the same procedure used for the Ku complex, except cleavage with PreScission Protease was performed using a 1:4 [w/w] ratio of PreScission Protease to Dnl4-Lif1 instead of the 1:5 [w/w] ratio used for the Ku complex.

**Nej1 purification.** *Saccharomyces cerevisiae* Nej1 was expressed in *Sf*21 insect cells (sourced from the European Molecular Biology Laboratory, Grenoble, France) with a 10xhis tag on its N-terminus using the Nej1_pFL vector[81]. The cell pellet was suspended in lysis buffer containing 25 mM Tris-HCl pH 8.0, 150 mM NaCl, 150 mM KCl, 10 mM β-mercaptoethanol, 50 mM imidazole, protease inhibitor cocktail

(cOmplete EDTA free, Roche, ½ tablet for 250 ml of lysis buffer) and sonicated by three cycles of 1 min at 60% amplitude. The lysate was supplemented with benzonase (0.04 U/ml) and 1 mM magnesium chloride, incubated for 15 min, and centrifuged at 50,000 × g for 30 min. The supernatant was incubated for 1 h at 4 °C under agitation with NiNTA resin pre-equilibrated with the lysis buffer. The resin was washed three times with 50 ml of washing buffer containing 25 mM Tris-HCl pH 8.0, 150 mM KCl, 850 mM NaCl, 10 mM β-mercaptoethanol, and 50 mM imidazole. The protein was eluted using elution buffer containing 25 mM Tris-HCl pH 8.0, 150 mM KCl, 150 mM NaCl, 10 mM β-mercaptoethanol, and 300 mM imidazole. The elution fractions were dialyzed at 4 °C against QA buffer containing 25 mM Tris-HCl pH 8.0, 100 mM KCl, 10 mM β-mercaptoethanol, and 10 mM EDTA using a 6−8 kDa cutoff dialysis membrane. The dialyzed sample was then loaded onto a 6 ml Resource Q column (Cytiva). The flow-through containing the protein of interest was dialyzed overnight at 4 °C against 1 l of storage buffer containing 25 mM Tris-HCl pH 8.0, 75 mM NaCl, 75 mM KCl, 10% glycerol, 5 mM β-mercaptoethanol and snap-frozen in liquid nitrogen.

**Yeast Exo1 purification.** Yeast Exo1 was purified from *Sf*9 cells using the pFB-Exo1-FLAG vector by FLAG affinity and HiTrap SP HP (Cytiva) ion exchange chromatography[77]. Yeast Rad54 and Pif1 were purified from *E. coli* cells using the pMALT-FLAG-Rad54 and pMALT-FLAG-Pif1 constructs by FLAG affinity purification[82].

**Sgs1, BLM, WRN, Mer3, and HLTF purification.** Sgs1, BLM, WRN, Mer3, and HLTF were expressed from *Sf*9 insect cells pFB-MBP-Sgs1, pFB1-MBP_BLM-wt_his, pFB1-MBP_WRN-wt_his, pFB_MBP_Mer3_his and pFastBac-2XMBP-hHLTFco-his vectors and purified by amylose affinity purification, followed by PreScission Protease removal of the N-terminal MBP tag (2xMBP for HLTF) and NiNTA affinity purification[83-86].

**Human PIF1 purification.** The sequence coding for human PIF1 was purchased from GenScript (see Supplementary Table 2 for synthetic gene sequences) and cloned into pFB-MBP-Sgs1-his using *Nhe*I and *Xma*I to generate pFB_MBP-hPIF1co_his. The sequence for the expression of human HELQ was amplified by PCR from the pFB-hHELQ-his[87] vector (a gift from Richard Wood) using NheI-hHELQ-FP and XmaI-hHELQ-RP (see Supplementary Table 1 for oligonucleotide sequences) and cloned into pFB1-MBP-WRN-wt_his via *Nhe*I and *Xma*I, using standard procedures to generate pFB-MBP-hHELQ-his. Human PIF1 and human HELQ were expressed in *Sf*9 cells and purified via amylose (New England Biolabs) and NiNTA (Qiagen) affinity chromatography as described for Sgs1. The MBP tag was removed by PreScission protease digestion.

**SMARCAL1, ZRANB3, HROB, and MCM8-MCM9 heterodimer purification.** SMARCAL1, ZRANB3, and HROB were purified by FLAG affinity chromatography from *Sf*9 insect cells with the pFB-FLAG-SMARCAL1, pFB-FLAG-ZRANB3-WT, and pFB-HROB-FLAG constructs[86,88]. The phosphorylated MCM8-MCM9 complex was obtained by co-expressing FLAG-tagged MCM8 and MBP-tagged MCM9 in the presence of 50 nM Okadaic acid with the pFB-FLAG-MCM8 and pFB-MBP-MCM9 plasmids and purified by amylose and FLAG affinity purification[88,89]. 50 nM Okadaic acid (APExBIO), 1 mM sodium orthovanadate (Sigma), 20 mM sodium fluoride (Sigma), and 15 mM sodium pyrophosphate (Applichem) were added to the lysis buffer to preserve the phosphorylation status[89].

**Human RAD54 purification.** The sequence coding for human RAD54 fused to a C-terminal FLAG-tag was codon optimized and purchased from GenScript (see Supplementary Table 2 for synthetic gene sequences) and cloned into pFB-MBP-Sgs1-his with *Nhe*I and *Hind*III,

generating pFB_MBP-TEV-hRAD54co-FLAG. A TEV cleavage site is present at the N-terminus of RAD54 for the removal of the MBP-tag. RAD54 was expressed in *Sf*9 cells with pFB_MBP-TEV-hRAD54co-FLAG. The protein was then purified by amylose and FLAG affinity purification as described for the Ku complex with some important modifications. First: the lysis buffer contained 5 mM β-mercaptoethanol instead of DTT. Second: after binding the protein on the amylose resin, a wash with wash buffer with 1 M NaCl was added. Third: the MBP-cleavage was performed using TEV protease (1:2.5, [w/w] ratio of TEV protease to protein) for 50 min at 4 °C, 10 min at room temperature, 50 min at 4 °C, and one last incubation for 10 min at room temperature. Fourth: the buffers used for the FLAG purification step contained 250 mM NaCl instead of 150 mM.

**FANCJ purification.** For the expression of FANCJ in *Sf*9 cells, the sequence of the protein was amplified from the BACH1-PVL1392 vector (a gift from Lumír Krejčí, Masaryk University)[90] using the FANCJ_to_pFB_for and FANCJ_to_pFB_rev oligonucleotides (see Supplementary Table 1 for oligonucleotide sequences), digested with *Bsa*I and inserted into pFB-Rad50co (digested with *Bam*HI and *Xho*I) to generate pFB-FLAG-FANCJ. The cell pellet was resuspended with 3 volumes of lysis buffer containing 50 mM Tris-HCl pH 7.5, 1 mM EDTA, 5 mM β-mercaptoethanol, 1 mM PMSF, 30 μg/ml leupeptin, and protease inhibitor cocktail (P8340, Sigma-Aldrich, 1:300) and let to swell on ice for 20 min. 50% glycerol and 5 M NaCl were added to a final concentration of 16.5% and 305 mM, respectively. Lysis was allowed to proceed for 30 min in agitation at 4 °C. The lysate was clarified by centrifugation at 50,000 × *g* for 30 min at 4 °C and incubated while mixing for 1 h with FLAG resin (Sigma). The resin was washed extensively with was buffer containing 50 mM Tris-HCl pH 7.5, 250 mM NaCl, 0.5 mM β-mercaptoethanol, 0.5 mM PMSF, 10% glycerol and 0.1% NP40. One last wash was performed with wash buffer without NP40, and the protein was eluted with wash buffer without NP40 supplemented with 5 μg/ml 3XFLAG peptide (GLPBIO). The fractions containing FANCJ were pooled, aliquoted, snap-frozen in liquid nitrogen, and stored at −80 °C until use.

**FANCM purification.** The coding sequence for FANCM was codon optimized for insect cell expression and purchased from Twist Biosciences in two separate synthetic genes termed FANCM_N and FANCM_C (see Supplementary Table 2 for synthetic gene sequences). The synthetic sequences were digested with *Nhe*I-*Afl*II (FANCM_N) and *Afl*II-*Hind*III (FANCM_C) and assembled into pFastBac-2XMBP-hHLTFco-his digested with *Nhe*I and *Hind*III to generate the pFastBac-2XMBP-hFANCMco-his vector. FANCM was expressed in *Sf*9 insect cells for 52 h. The cells were pelleted, washed with 1× PBS, snap-frozen, and stored at −80 °C until further use. For the purification, which was carried out entirely at 4 °C or on ice, the pellet was resuspended in three volumes of lysis buffer containing 50 mM Tris-HCl pH 7.5, 1 mM EDTA, 5 mM β-mercaptoethanol, 2 mM PMSF, 50 μg/ml leupeptin and protease inhibitor cocktail (P8340, Sigma-Aldrich, 1:200) and incubated in agitation for 5 min at 4 °C. After incubation, one-half volume of 50% glycerol was added to the mixture, followed by 6.5% volume of 5 M NaCl (final concentration 305 mM) and extraction was carried out for 25 min at 4 °C. The soluble extract was obtained by centrifugation of the extract at 50,000 × *g* for 30 min at 4 °C and incubated in agitation for 1 h with amylose resin (New England Biolabs) pre-equilibrated with lysis buffer supplemented with glycerol and NaCl. The resin was washed 2 times batchwise and then extensively on a disposable plastic column (Pierce) with wash buffer containing 50 mM Tris-HCl pH 7.5, 1 mM EDTA, 5 mM β-mercaptoethanol, 10% glycerol, 1 mM PMSF, and 1 M NaCl. The concentration of salt was progressively lowered by washing the resin with wash buffer containing 300 and 150 mM NaCl (one column volume each). The protein was eluted with 150 mM wash buffer supplemented with 10 mM maltose.

The eluted protein was quantitated by Bradford assay and incubated with a 1:5 [w/w] ratio of PreScission Protease to protein for 1 h at 4 °C to remove the 2xMBP-tag. After the incubation, the protein was diluted with buffer R containing 50 mM Tris-HCl pH 7.5, 5 mM β-mercaptoethanol, 10% glycerol, and 1 mM PMSF to lower the salt to 100 mM and loaded onto a 1 ml HiTrap Q HP column (Cytiva). The protein was washed with 20 ml buffer R at 100 mM NaCl and eluted from the column with a salt gradient in buffer R from 100 mM to 1 M NaCl. The fractions containing the protein were pooled, aliquoted, and snap-frozen.

**HLTF variants cloning and purification.** The HLTF N90A-N91A (NANA) was purified as the WT protein using pFastBac-2XMBP-hHLTFco-his-NANA. The expression vector was generated from the pFastBac-2XMBP-hHLTFco-his vector by mutagenesis using QuikChange II XL Site-Directed Mutagenesis Kit (Agilent) according to the manufacturer's instructions using HLTF_NANA_for/HLTF_NANA_rev (see Supplementary Table 1 for primer sequences). For the expression of the HLTF HIRAN domain variants (WT and NANA), the first 180 amino acids of HLTF were amplified from the pFastBac-2XMBP-hHLTFco-his vector (WT and NANA) by PCR using the primers HLTF_1-180_for and HLTF_1-180_rev (see Supplementary Table 1 for primer sequences) and inserted in the pFastBac-2XMBP-HLTF-his in place of the full-length protein, creating pFastBac-2XMBP-HIRAN-his and pFastBac-2XMBP-HIRAN-his-NANA. Expression and purification were performed following the same procedure used for the full-length proteins except the MBP tag was not removed.

**Human and yeast RPA purification.** Human RPA was expressed in *Sf*9 insect cells with the pFB-RPA1, pFB-RPA2, pFB-6xhis-RPA3 vectors and purified by NiNTA affinity chromatography, on a HiTrap Blue column (Cytiva) followed by desalting with a HiTrap Desalting column (Cytiva) and finally onto a HiTrap Q column (Cytiva)[91–93]. Yeast RPA was expressed in BL21 (DE3) pLysS cells and purified on a HiTrap Blue column (Cytiva), HiTrap Desalting column (Cytiva) and a HiTrap Q column (Cytiva)[91].

Srs2 and Mph1 were gifts from Lumír Krejčí (Masaryk University)[94,95]. RECQ5 and FBH1 were provided by Pavel Janscak (University of Zürich)[96,97]. Chd1 was obtained from Beat Fierz (EPFL, Lausanne)[98]. The FACT complex (hSpt16-SSRP1) was a kind gift from Jacob Corn (ETH Zürich)[99].

## Cas9 and Cas12a RNP generation

crRNA (CD.Cas9.GJXV6830.AD) and tracrRNA for Cas9 targeting were purchased from Integrated DNA Technologies (see Supplementary Table 1 for oligonucleotide sequences) and annealed at equimolar concentrations (10 μM final) in IDTE buffer pH 8.0 (Integrated DNA Technologies) according to manufacturer's instructions. The Cas12a crRNA (pUC19_Cas12a_1, see Supplementary Table 1 for sequence) was used without further modifications. 1 μM (final) Cas9 and Cas12a variants were incubated with a three-fold excess of the RNA component (annealed crRNA-tracrRNA for Cas9 and crRNA for Cas12a) in RNP buffer containing 25 mM Tris-HCl pH 7.5, 5 mM magnesium chloride, 1 mM dithiothreitol, 0.25 mg/ml bovine serum albumin, BSA and 150 mM KCl for 10 min at 25 °C to produce the respective RNPs. After incubation, the RNPs were subaliquoted, snap-frozen in liquid nitrogen, and stored at −80 °C for later use.

## Preparation of plasmid-based substrates

The plasmids used for the in vitro assays are derivatives of pUC19 (see Supplementary Table 1 for a list of plasmids used in this study). The pUC19_0x_LacO plasmid was obtained by inserting the annealed oligonucleotides Even_A and Even_B into pUC19 digested with *Hind*III and *Afl*III (see Supplementary Table 1 for oligonucleotide sequences). pUC19_1x_LacO (used for the experiments in Fig. 2a–c) was produced

inserting the oligonucleotides Even(WT)_1 and Even(WT)_2 into the *Bbs*I site of pUC19_0x_LacO. The linear, nicked circular 1, and nicked circular 2 versions of the substrate used in Fig. 2a–c were obtained by digestion of pUC19_1x_LacO with *Eco*RV (New England Biolabs), Nt.*Bst*NBI and Nb.*Bbv*CI, respectively. The substrates with open- and loop-end used for experiments in Fig. 2d–f and Supplementary Fig. 2b were produced from the pAttP-S plasmid using the ΦC31 integrase with Open_int_LacO_TOP annealed to Open_int_LacO_BOT, or with self-annealed Hairpin_integrase_LacO, respectively (see Supplementary Table 1 for oligonucleotide sequences)[92].

The pUC19_0xLacO_Cas substrate used for Cas9 and Cas12a nuclease experiments was obtained by site-directed mutagenesis (QuikChange II XL Site-Directed Mutagenesis Kit, Agilent) with pUC19_0xLacO_Cas_FOR and pUC19_0xLacO_Cas_REV primers (see Supplementary Table 1 for oligonucleotide sequences). The substrate with the inverted target guide site used in Supplementary Fig. 3a (pUC19_0xLacO_Cas_rev) was obtained by replacing the portion of pUC19_0xLacO_Cas between *Hind*III and *Pst*I sites with annealed oligonucleotides pUC19_0xLacO_Cas_rev_oligo1 and pUC19_0xLacO_Cas_rev_oligo2. The substrate for the in vitro ligation assay was prepared by PCR from the pUC19_0xLacO_Cas plasmid with primers pUC19_0xLacO_Cas_1kb_for and pUC19_0xLacO_Cas_1kb_rev, according to standard procedures (see Supplementary Table 1 for oligonucleotide sequences). [α-$^{32}$P]dCTP (Hartmann Analytic) was added to the PCR reaction in order to label the substrate radioactively.

## Preparation of oligonucleotide-based substrates

The oligonucleotide-based substrate for the assay presented in Fig. 3c was prepared by labeling Cas9_target1_FO oligonucleotide (see Supplementary Table 1 for oligonucleotide sequence) at the 5′-end with T4 polynucleotide kinase (New England Biolabs) and [γ-$^{32}$P]ATP (Hartmann Analytic). After labeling, the oligonucleotide was purified on a Micro Bio-Spin P-30 Gel Column (Bio-Rad). The labeled oligonucleotide was then annealed with a 2-fold excess of Cas9_target1_RE oligonucleotide (see Supplementary Table 1 for oligonucleotide sequence) in annealing buffer (10 mM Tris-HCl pH 8, 50 mM NaCl, 10 mM magnesium chloride), heated to 95 °C for 3 min and cooled down to room temperature overnight. The oligonucleotide-based substrate for the assay presented in Supplementary Fig. 7f was prepared by labeling Cas9-TOP (see Supplementary Table 1 for oligonucleotide sequence) at the 5′-end and annealed with a 2-fold excess of Cas9-BOT (see Supplementary Table 1 for oligonucleotide sequence) as described above.

The mobile Holliday junction substrate (see Supplementary Table 1 for oligonucleotide sequences)[86] was prepared by annealing 3′-labeled XO2 with XO1 (1:1.2 ratio) in annealing buffer (10 mM Tris-HCl pH 8, 50 mM NaCl, 10 mM magnesium chloride) by heating for 3 min at 95 °C and slowly cooling down the mix to room temperature overnight. At the same time, XO1c.MM2 and XO2c.MM oligonucleotides (1.2:1.2 ratio compared to the labeled oligonucleotide) were annealed as described above. The next day, the mixes were combined and incubated for 30 min at 37 °C and cooled down to room temperature for 2 h. XO2 was 3′-end labeled using Terminal Transferase (New England Biolabs) and [α-$^{32}$P]dCTP (Hartmann Analytic) according to the manufacturer's instructions and purified on Micro Bio-Spin P-30 Gel Columns (Bio-Rad).

## Electrophoretic mobility shift assays

DNA binding experiments with the MR complex and the various substrates in Supplementary Fig. 2a were performed in binding buffer containing 25 mM Tris-acetate pH 7.5, 1 mM dithiothreitol, 5 mM magnesium chloride, 1 mM ATP-γ-S (BIOLOG), 0.25 mg/ml bovine serum albumin (New England Biolabs) and 1 nM substrate, in a final volume of 15 µl. The MR complex was added at the indicated final concentration to the mix and incubated for 15 min a 30 °C. After the

incubation, 5 µl EMSA loading dye (50% glycerol with bromophenol blue) was added to the reaction, which was loaded on a 0.6% agarose gel and run at 4 °C. At the end of the run, the gel was stained with GelRed (Biotium) and imaged with a Typhoon Imager (Cytiva).

DNA binding experiments with Cas9 and the HIRAN variants in Supplementary Fig. 7f were performed in binding buffer containing 25 mM Tris-acetate pH 7.5, 1 mM dithiothreitol, 5 mM magnesium chloride, 0.25 mg/ml bovine serum albumin (New England Biolabs), 50 mM potassium chloride and 1 nM substrate, in a final volume of 15 µl. The substrate was incubated with 5 nM Cas9 HA or DA for 30 min a 30 °C, as indicated. Reactions were moved on ice, supplemented with the indicated concentration of HLTF HIRAN domain WT or NANA mutant, and incubated for 10 min at 37 °C. After the incubation, 5 µl EMSA loading dye (50% glycerol with bromophenol blue) was added to the reaction. The DNA-protein complexes were separated on a 0.6% agarose gel and run at 4 °C. The gel was dried on DE81 chromatography paper (Whatman), exposed to phosphor screens, imaged with a Typhoon Imager, and quantitated using ImageJ Software (1.53 g). Plots were generated with Graph Pad Prism 10.2.2 (397).

## Exo1 nuclease assay

Nuclease assays with Exo1 were performed in reaction buffer containing 1 nM (in molecules) of DNA substrate, 25 mM Tris-acetate pH 7.5, 1 mM dithiothreitol, 5 mM magnesium chloride, 1 mM manganese chloride, 1 mM ATP, 80 U/ml pyruvate kinase (Sigma), 1 mM phosphoenolpyruvate, 50 mM KCl and 0.25 mg/ml bovine serum albumin (New England Biolabs), in a final volume of 15 µl. For experiments in Supplementary Fig. 6, reactions contained 237 nM yeast RPA. The substrate was incubated with the indicated amounts of *Eco*RV, Cas12a, or Cas9 variants for 10 min at 37 °C. Reactions were moved on ice, supplemented with Exo1 and HLTF, as indicated and incubated for 30 min at 30 °C. The samples were supplemented with 5 µl of 0.2% STOP solution (150 mM EDTA, 0.2% SDS, 30% glycerol, and 1 mg/ml bromophenol blue) and deproteinated with 1 µl of 14–22 mg/ml Proteinase K (Roche) for 1 h at 37 °C. The stopped reactions were separated on 1% agarose gels stained with GelRed (Biotium), imaged with a Typhoon Imager (Cytiva), and quantitated using ImageJ Software.

## Annealing DNA end resection assays

Annealing DNA end resection assays with MR-pSae2 were performed in reaction buffer containing 1 nM (in molecules) of the indicated DNA substrate, 25 mM Tris-acetate pH 7.5, 1 mM dithiothreitol, 5 mM magnesium chloride, 1 mM manganese chloride, 1 mM ATP, 80 U/ml pyruvate kinase (Sigma), 1 mM phosphoenolpyruvate, 50 mM KCl (omitted for experiments in Fig. 2) and 0.25 mg/ml bovine serum albumin (New England Biolabs), in a final volume of 15 µl. For experiments in the presence of LacI, the substrate was incubated with 84 nM LacI (in monomers) for 5 min at 30 °C. The reaction was then moved on ice. Unless indicated otherwise, 25 nM Mre11-Rad50 (MR) or Mre11-Rad50-Xrs2 (MRX) complexes and 200 nM phosphorylated Sae2 (pSae2) were added, and the reactions were further incubated for 30 min at 30 °C. The reactions with human MRE11-RAD50-NBS1 (MRN, 40 nM) complex and phosphorylated CtIP (pCtIP, 100 nM) were incubated at 37 °C for 1 h, unless stated otherwise. The reactions were stopped with 5 µl of 2% STOP solution (30 mM EDTA, 2% SDS, 30% glycerol, and 1 mg/ml bromophenol blue) and 1 µl of 14–22 mg/ml Proteinase K (Roche). Deproteination was carried out for 1 h at 37 °C. Subsequently, the stopped reaction was supplemented with 6.5 mM final magnesium acetate and a three-fold excess of the radioactive probe (Probe 1 except for the left panel in Fig. 3a and Supplementary Fig. 3a for which Probe 2 was used, see Supplementary Table 1 for the sequences), heated to 60 °C for 3 min and slowly cooled overnight to let the probe anneal to the resected DNA. The probe was previously radioactively labeled at the 5′-end using T4 polynucleotide kinase (New

England Biolabs) and [γ-$^{32}$P]ATP (Hartmann Analytic) and purified on Micro Bio-Spin P-30 Gel Columns (Bio-Rad). The annealing reactions were separated on 1% agarose gels that were then dried on DE81 chromatography paper (Whatman). The dried gels were exposed to phosphor screens, imaged with a Typhoon Imager, and quantitated using ImageJ Software.

Unless indicated otherwise, for resection of breaks generated by *EcoR*V, Cas9, or Cas12a, the substrate was incubated with the indicated nuclease(s) for 10 min at 37 °C before the addition of the DNA end resection components. Assays as shown in Fig. 1b were carried out as described above with the following modifications: reactions were stopped after incubation with the Cas9 and Cas12a variants and separated on 1% agarose gels stained with GelRed (Biotium) and imaged with a Typhoon Imager (Cytiva).

For resection experiments performed in the presence of translocases, the indicated proteins were added together with the resection components and incubated as described above. The assays in Supplementary Fig. 4d were performed as described above with a few important modifications. Magnesium chloride was added at a final concentration of 1 mM while manganese chloride was omitted, the indicated concentrations of FANCM and HLTF were incubated with the substrate for 30 min at 37 °C. Additional 4 mM of magnesium chloride and 1 mM manganese chloride were added together with the resection components, and the reactions were performed for 1 h at 30 °C instead of 30 min.

Resection assays carried out in the presence of purified HIRAN domain were supplemented with the HLTF fragment for 10 min at 37 °C before the addition of full-length HLTF and the resection components.

## Multi-turnover assays

Nuclease assays for detection of multi-turnover behavior of Cas9 variants were performed in 15 μl in reaction buffer containing 1 nM substrate (in molecules), 25 mM Tris-acetate pH 7.5, 1 mM dithiothreitol, 5 mM magnesium chloride, 1 mM manganese chloride, 1 mM ATP, 80 U/ml pyruvate kinase (Sigma), 1 mM phosphoenolpyruvate, 50 mM KCl and 0.25 mg/ml bovine serum albumin (New England Biolabs). The substrate was incubated with the indicated amount of Cas9 variant for 10 min at 37 °C. Subsequently, the reaction was supplemented with the indicated amount of the HLTF variant and further incubated for 30 min at 37 °C (2 h for the down-titration of Cas9, Fig. 5c and Supplementary Fig. 5b). For experiments performed in the presence of the HIRAN domain, the indicated amount of the HLTF fragment was incubated with the Cas9-treated substrate for 10 min at 37 °C before the addition of full-length HLTF.

## Non-homologous end-joining assays

In vitro ligation assays were performed in two steps. First, 1 nM plasmid-length substrate was incubated for 10 min at 37 °C with 0.4 U of *Eco*RV or 5 nM Cas9 in ligation buffer containing 60 mM Tris-HCl pH 7.5, 5 mM dithiothreitol, 10 mM magnesium chloride, 1 mM ATP, 0.05 mg/ml bovine serum albumin (New England Biolabs), in a final volume of 3 μl. For the second step, the reaction was then supplemented with ligation buffer to a volume of 13 μl to lower the substrate concentration to 0.2 nM. 9 nM Ku complex, and the remaining components of the yeast NHEJ machinery at the indicated concentrations were subsequently added to the reaction, and the samples were incubated for 1 h at 25 °C. Reactions were deproteinated at 50 °C for 1 h after the addition of 5 μl of 2% STOP solution (30 mM EDTA, 2% SDS, 30% glycerol, and 1 mg/ml bromophenol blue) and 1 μl of 14−22 mg/ml Proteinase K (Roche). Samples were separated on a 1.5% agarose gel at 120 V, the gel was then dried on DE81 chromatography paper (Whatman), exposed to phosphor screens and imaged with a Typhoon imager. Ligation efficiency was quantitated using ImageJ Software. For the experiments in Fig. 4e, the substrate was treated with the indicated amount of HLTF for 10 min at 37 °C before the addition of the NHEJ components.

## Branch-migration assays

Branch-migration assays were performed in a volume of 15 μl in reaction buffer containing 25 mM Tris-acetate pH 7.5, 1 mM dithiothreitol, 5 mM magnesium chloride, 1 mM manganese chloride, 1 mM ATP, 80 U/ml pyruvate kinase (Sigma), 1 mM phosphoenolpyruvate, 0.25 mg/ml bovine serum albumin (New England Biolabs) and 1 nM mobile Holliday Junction substrate (refer to the section Preparation of oligonucleotide-based substrates)[86]. After the addition of the indicated proteins, the reaction was incubated for 30 min at 30 °C. Branch migration was stopped by the addition 5 μl of 2% STOP Solution and 1 μl of Proteinase K and incubation for 10 min at 37 °C. Reaction products were separated on 8% TBE (89 mM Tris, 89 mM boric acid, 2 mM EDTA) acrylamide gels. After the run, gels were dried on 17CHR paper (Whatman) and treated as described above.

## Nucleic acid and protein preparation for single-molecule fluorescence experiments

The dsDNA target was prepared by mixing oligonucleotides TS, NTS, and 22nt-adapter (purchased from Integrated DNA Technologies, see Supplementary Table 1 for oligonucleotide sequences) at 1:1.25:1 ratio in a buffer containing 10 mM Tris-HCl pH 8 and 50 mM NaCl. The oligonucleotides were heated at 95 °C for 1 min, and slowly cooled down to room temperature over 1 h[64]. For experiments with Alexa750 on the PAM-proximal side of the target DNA, TS was substituted with TS-1 (see Supplementary Table 1 for oligonucleotide sequences). wtCas9 and dCas9 were gifts from the Doudna lab at UC Berkeley. Cas9 H840A nickase was purchased from Applied Biological Materials Inc. (K136). Cas9 D10A nickase was purchased from NEB (M0650T). ATTO550 labeled tracrRNA was purchased from IDT. crRNA was transcribed in vitro using HiScribe™ T7 Quick High Yield RNA Synthesis Kit (NEB, E2050S). TracrRNA and crRNA were annealed to assemble gRNA following manufacturer's instructions[64].

## Single-molecule experiments

DNA substrates were added into the sample chamber of a quartz microscope slide and immobilized on the surface of the microscope slide via biotin-NeutrAvidin interactions for TIRFM imaging[100]. The TIRFM sample chamber was maintained at 37 °C throughout the experiment. To assemble Cas9 RNP, 300 nM Cas9 and 100 nM gRNA were mixed in Cas9 imaging buffer (20 mM Tris-HCl pH 8, 100 mM KCl, 5 mM magnesium chloride, 5% [vol/vol] glycerol, 0.2 mg/ml bovine serum albumin (New England Biolabs) and saturated Trolox (>5 mM), 0.8% (w/v) dextrose, 1 mg/ml glucose oxidase, 0.04 mg/ml catalase), and incubated for 10 min at room temperature. Next, the assembled Cas9 RNP was diluted to 20 nM in Cas9 imaging buffer, added into the sample chamber, and incubated for 10 min. After flowing out free Cas9 RNP from the sample chamber using Cas9 imaging buffer, fluorescence signals of more than 1000 molecules were measured to make the "Before HLTF" histograms. Next, 20 nM HLTF in the HLTF reaction buffer (25 mM Tris-HCl pH 7.5, 1 mM dithiothreitol, 5 mM magnesium chloride, 1 mM manganese chloride, 1 mM ATP, 80 U/ml pyruvate kinase [Sigma], 1 mM phosphoenolpyruvate and 0.25 mg/ml bovine serum albumin (New England Biolabs) and 50 mM KCl) was added into the sample chamber and incubated for 30 min (unless otherwise indicated). The HLTF was then flowed out of the sample chamber using Cas9 imaging buffer, and fluorescence signals of >1000 molecules were measured for making the "After HLTF" histograms. For experiments with two fluorophores, the signal intensity ratio is defined as the intensity of Cy5 divided by the total intensity of Cy5 and ATTO550. For experiments with three fluorophores, the signal intensity ratios E21 and E23 are defined as the intensity of ATTO550 divided by the total intensity

of Cy5 and ATTO550 (Eq. 1) and the intensity of Alexa750 divided by the total intensity of Cy5 and Alexa750 (Eq. 2), respectively.

$$E21 = \frac{I_{ATTO550}}{I_{ATTO550} + I_{Cy5}} \qquad (1)$$

$$E23 = \frac{I_{Alexa750}}{I_{Alexa750} + I_{Cy5}} \qquad (2)$$

## MCF10A cell culture, treatments, and western blotting

MCF10A cells (CRL-10317, ATCC) were cultured in a 1:1 mixture of DMEM and Ham's F12 medium (Thermo Fisher Scientific), supplemented with 20 ng/ml human epidermal growth factor (Sigma-Aldrich), 10 μg/ml insulin (Sigma-Aldrich), 0.5 μg/ml hydrocortisone (Sigma-Aldrich), 100 ng/ml cholera toxin (Sigma-Aldrich), 5% horse serum (Thermo Fisher Scientific), and Pen-Strep (Gibco). MCF10A DOX inducible nCas9D10A and nCas9H840A cells were constructed infecting MCF10A cells with lentivirus containing Dox-Cas9-D10A and Dox-Cas9-H840A and selected with 10 μg/ml blasticidin[101]. The guides that target Alu (CAGGCGTGAGCCACCGCGCC) and LINE1 (ATTCTACCAGAGGTACAAGG) elements[67] were cloned into the Lenti-Guide-NLS-GFP[101] and delivered by lentivirus. After virus infection, MCF10A cells were selected using 3 μg/ml puromycin. For the Western blotting assay, nCas9D10A or nCas9H840A MCF10A cells were treated or not with siRNA targeting HLTF (L-006448-00-0005 Dharmacon Smart pool) for 48 h. Doxycycline was added to the media for 24 h before collection to induce the nickase expression. To obtain whole cell extracts, cells were lysed in Laemmli lysis buffer (4% SDS, 20% glycerol, 125 mM Tris-HCl pH 7.4, 50 mM b-Glycerophosphate disodium, 2 mM PMSF, and 1× Complete Mini EDTA free proteinase inhibitor [Roche]), boiled for 5 min, sonicated for 10 s at amplitude 20% in an Ultrasonic Processor (Cole-Parmer), and centrifuged at 16,000 × g for 10 min. The supernatant was then collected, and 40 μg of total protein was loaded per well in 4–12% Bis-Tris gels (NuPAGE, Invitrogen) and transferred onto nitrocellulose membranes. Primary antibodies were used at the following dilutions: anti-Cas9 (1:1000, Novus Biologicals, NBP280679), anti-HLTF (1:500, Abcam, ab17984), anti-γH2AX (1:5000, pSer 139, Millipore γH2AX (1:5000, pSer 139, Millipore 05-636), anti-pRPA S4/S8 (1:1000, Bethyl Laboratories, Cat# A300-245A), anti-Tubulin (1:5000, Sigma, #T-5168). Except for anti-γH2AX, which was incubated for 1 h at room temperature, all other primary antibodies were incubated overnight at 4 °C. Fluorescent secondary antibody anti-mouse or anti-rabbit IRDye 800CW or 680RD (LI-COR Biosciences) were incubated for 1 h at room temperature. Detection of protein bands was performed by fluorescence imaging using a Li-Cor Odyssey CLx imaging system (Li-Cor Biosciences).

## Statistics and reproducibility

The number of repeats was chosen based on what is common and/or practical in the field. Data points were excluded only when there was a valid reason to do so, such as errors during the assay execution and/or imaging, or failure of experimental apparatuses (e.g. ruined gel wells). The experiments were repeated multiple times, as indicated in the figure legend. The indicated number of repeats in the figure legends refers to biological replicates. Randomization and blinding were prevented by the loading order of the sample. This was nevertheless not a problem as quantification was carried out objectively with dedicated software.

## Reporting summary

Further information on research design is available in the Nature Portfolio Reporting Summary linked to this article.

## Data availability

No large datasets were generated or analyzed during the current study. All the data supporting the findings of this study are available within the paper and its supplementary information files. Source data are provided with this paper.

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

## Acknowledgements

We thank the members of the Cejka laboratory for their critical comments on the manuscript and Damiano Borrello and Lepakshi Ranjha for providing purified proteins. We thank P. Janscak (University of Zürich), T. Paull (University of Texas), L. Krejci (Masaryk University, Brno), B. Fierz (EPFL, Lausanne), J. Doudna (UC Berkeley) and J. Corn (ETH Zürich) for protein expression constructs or purified proteins. Research in the Cejka laboratory is funded by the Swiss National Science Foundation (SNSF) (Grants 310030_207588 and 310030_205199 to P.C.) and the European Research Council (ERC) (Grant 101018257 to P.C.). M.R.D.S. is a recipient of EMBO long-term postdoctoral fellowship (ALTF 710-2021 to M.R.D.S.). The US National Institutes of Health (R35 GM 122569 to T.H.) supported research in the Ha laboratory. The Agence nationale de la recherché (Grants ANR-22-CE12-0037, ANR 23-CE11-0033 and ANR-10-INBS-0005 to J.P.C.) supports the research in the Charbonnier laboratory. T. Ha is an investigator with the Howard Hughes Medical Institute.

## Author contributions

G.R. and P.C. conceived the study. G.R. designed, performed, and analyzed the ensemble biochemical experiments with the help of M.R.D.S. Y.W. and J.H. designed, performed and analyzed the single-molecule fluorescence experiments under the supervision of T.H. R.P. designed, performed and analyzed the cellular experiments under the supervision of A.N. E.C. expressed and purified Dnl4-Lif1 complex. I.C. expressed and purified human BLM, WRN, the MRN complex and phosphorylated CtIP. S.H. initially expressed and purified HLTF. S.B. expressed and purified Exo1, yeast RPA and human RPA. A.A. expressed and purified MCM8-MCM9 and HROB, and cloned the constructs for the expression of human RPA. M.R.D.S. additionally expressed and purified FANCM and HLTF NANA mutant. V.M. and V.R. expressed and purified the Nej1 protein under the supervision of J.B.C. M.S. purified the Cas9 and Cas12a variants under the supervision of M.J. G.R., and P.C. wrote the manuscripts with inputs from all authors. P.C., T.H., A.N., M.J., and J.B.C. acquired the funding for the project.

## Competing interests

The authors declare no competing interests.
