## [Peer Review File · Nature Communications]

HLTF disrupts Cas9-DNA post-cleavage complexes to allow DNA break processingREVIEWER COMMENTS

Reviewer #1 (Remarks to the Author):

The Cejka laboratory has investigated mechanisms that regulate the efficiency of repair of DNA double-strand breaks introduced by Cas9. By biochemical means, the authors have found that the Cas9 breaks are poorly resected by the Mre11-Rad50-pSae2 or Mre11-Rad50-Xrs2-pSae2 complex. Additional experiments help establish that Cas9-introduced breaks are more refractory to end resection enzymes because of end bridging by Cas9. The authors further demonstrate by single-molecule and bulk biochemical analyses that the motor protein HLTF can remove Cas9 from DNA to facilitate resection and hence DSB repair.

The data presented are of high quality and cogently explain why DNA double-strand breaks introduced by Cas9 are repaired more slowly than other breaks, and they also implicate HLTF in the conversion of Cas9-associated breaks into a structure amenable to DNA end processing and repair. This is a very nice study that is suited to Nature Communications pending revisions.

Major Comments:

1. Is there direct evidence that DNA ends stay bridged by Cas9 in the biochemical analyses?
2. Why aren't degradation products generated by EXO1 detected in Figure 1d.
3. Rad5 is the yeast ortholog of human HLTF. It is not clear why it was not tested along with other yeast proteins.
4. The model in Figure 6K suggests that HLTF interacts with 3'overhangs generated by Cas9 activity. Does HLTF requires Cas9 for its recruitment? Have the authors checked if HLTF but not HLTF NANA is recruited to DNA after Cas9 has acted? Do Cas9 and HLTF physically interact, perhaps through HIRAN of the latter?
5. Resection of Cas9 breaks occurs upon addition of the yeast helicases Srs2 and Mph1 in ED Figure 3a. This result should be discussed.
6. Is MRX more, or less, efficacious than MRN in cleaving the Cas9 intermediate? The authors should provide quantitation to answer this question.

Minor Comments:

1. Figure 1C: Some resection is seen in the presence of Cas9 (lane 4). Authors should modify the statement "...while DNA cut by Cas9 was not processed".
2. Have the authors checked if cleavage products generated by Cas12a are a suitable NHEJ substrate?
3. Figure 4d: degradation products are not visible.
4. Have authors tried to perform NHEJ assays in the presence of higher HLTF concentrations (Figure 4f)?
5. The NANA mutation in HLTF reduces the resection efficiency by 50% compared to WT HLTF (Figure 6F). Discuss this thoughtfully, as it suggests that another HLTF region is also involved in Cas9 removal.
6. Figure 2, C and F: Some of the error bars are compromised; the y-axis needs to be adjusted to fix this.

Reviewer #2 (Remarks to the Author):

In this manuscript, Reginato et al. addressed two key questions: Firstly, the Cas9 residing on cleaved DNA conceals the DSB from access by repair factors and delays the repair process, e.g. end process initiated by MRN complex (MRX in yeast). Exposure of loose DSB end activates MRN which endonucleolytically cleaves 5'-terminated DNA strand even with the protein blocks, such as dCas9, bound adjacent to the DNA end. Then backward resection in a 3' to 5' direction by MRN removes the

protein blocks and creates the 3' terminals. While the Cas12a releases the PAM-distal end of DNA and initiates MRN-mediated resection immediately. This unique behavior of Cas9, and consequently impact on repair are well known. Secondly, it provides strong in vitro evidence that HLF1 removes DNA bound Cas9 (or H840A) and converts Cas9 or H840A into multi-turnover enzymes. Removal of Cas9 or H840A initiates subsequent DDR and repair. Mechanistically, HLF1 used the HIRAN domain to recognize the free 3' non-target ssDNA released from the Cas9-sgRNA-DNA complex. Therefore, HLF1 only dislocates H840A nickase but not D10A. These findings revealed a new regulator in controlling the target residence of Cas9, in addition to DNA metabolism events including transcription, replication and remodeling. This study not only provides valuable insights into the understanding of DNA damage and repair when Cas9 is used to introduce the breaks, but also be very helpful to the gene editing community, especially for the Prime editor system developed on H840A. Meanwhile I notice a paper (ref 14) that has also demonstrated the HLF1 function in release of H840A, but I am unable to access the original manuscript since this work has not been published yet. Taken together, the data are convincing and support their conclusions. I do have several concerns before the acceptance.

1. In the MR-pSae2 resection assays. To active the MR complex, it requires a 3 to 8-fold excess of pCtIP (p-Sae2) over MR? For Fig1a. the radioactive probe was labeled at a distance about 90nt away to the Cas9 bound end, but in ref50, the distance is 160nt. The probes were designed to move toward the DNA end due to the bound Cas9 on DNA?
2. In Fig1C, WT Cas9 cutting demonstrated a very weak band showing limited resection by MR-pSae. Because the reaction was performed for a short time, it is possible that more Cas9 would dissociate from cleaved DNA spontaneously and promote the end resection by MRN with longer incubation time (more than 10h)?
3. In Fig2b, the schematic representation of DNA substrate did not be correctly described in the text? Nicked circular 1 indicates the substrate with LacI block placed 0.9kb away from the nick, and Nicked circular 2 for the substrate with LacI block placed adjacent to the nick.
4. In Fig2c. Nicked circular 2, with blocks placed adjacent to nick, could be processed by MR-pSae, albeit to a much less extent. The MRN can somehow sense the nick end and initiate resection?
5. Fig3A. The resection in the case of EcoRV+Cas9 (lane 5 or 10) is much stronger than that of EcoRV cutting along (lane 3 or 8), because the Cas9 block stimulates the MRX endonuclease activity. Post EcoRV cleavage, The Cas9 bound on one end can stimulate the MRN-mediated resection of DNA on the other side?
6. Fig3C. lane4. The band at 30nt indicates the products which MR-pSae incises the Ku-bound DNA 30nt away from the end. A weaker band at 60nt suggests another incision point of Mre11? Whether it occurs on dCas9 bound DNA?
7. Fig3F. With an excess of Cas9 (5nM) over DNA (1nM), there is still considerable level of uncut substrate. If these DNA could be ligated, the ligation products would be observed as larger bands on the gels, similar to those present in lane3, in which EcoRV cleaved DNA substrates could be ligated to form the large DNA products.
8. Fig4d. HLF1 dislocates the Cas9 and exposes the DNA for Exo1 degradation. Whether Exo1 recognize the nicked DNA and initiate resection when HLF1 dislocate H840A?
9. Follow the above question. In Line 152. The authors indicated "Such DNA end sensing capacity prevents spurious cleavage of genomic DNA away from DSB sites". Based on the results, the MRN would not initiate resection from the nick without a loose DSB end. In other words, the Mre11 would not process the DNA nicks or gaps in genome without a DSB present adjacent to them? Recently, a study (Nature Communications (2023) 14:6265) showed the Mre11 and Exo1 bidirectionally process the replication-derived ssDNA gap and create a DSB when the gap is extended. It suggests the distinct mechanism of MRN resection in mammalian cells.
10. The Cas12a resides on one end of the DNA break. How does Cas12a dissociate from DNA. The dissociation of Cas12a from DNA requires a specific translocase?

Reviewer #3 (Remarks to the Author):

This manuscript by Reginato and colleagues beautifully show that in vitro, Cas9 breaks cannot be resected by the MRE11 complex. They furthermore show that to allow resection, Cas9 is required to be dissociated by the translocase HLTF which directly removes Cas9 from the broken ends which then allows for further processing. The authors also show that the removal of Cas9 is mediated by the binding of HIRAN domain of HLTF and also the release of 3' end by Cas9. This is an important study, which mechanistically elucidates why Cas9 generated breaks takes time to be repaired. The work has been well designed and provides critical mechanistic insights for the mechanism of Cas9 mediated DSB repair. Below are some comments which need to be addressed before the manuscript is published.

Comments:

1. It is unclear to me why the looped DNA substrate (fig2.d and e) is resected by MR-pSae2. From fig2.b it is clear that probably a double stranded end with the protein block is required for resection. However, the looped substrate wouldn't resemble a double stranded end. How is then the break sensed to initiate resection?
2. In fig3.c the authors show that using Ku or dCas9 the incision point by MR-pSae2 is dependent on the footprint of the protein block used. Would they also observe similar resection if an active Cas9 were to be used instead of dCas9 on the same substrate?
3. In fig.4b the authors show that FANCM also removes Cas9 and promotes resection. This is an interesting observation which was not followed up on. Is the mechanism of FANCM removal of Cas9 similar to the one observed by HLTF. The authors need to parse this out a bit more. Furthermore quantitation of FANCM mediated removal of Cas9 is missing in comparison to HLTF. Also, FANCM removal of Cas9 in combination with MRN-CtIP should also be shown along with MR-pSae2.
4. I am a bit confused with fig.3c and extended fig.3f. In fig3.c the authors show that use of dCas9 can have incision by MR-pSae2. However, in extended fig.3f, the authors show that HLTF addition of HLTF to dCas9 does not induce resection. Is it because HLTF cannot bind to the 3' end generated by active Cas9? Can addition of FANCM to this reaction induce resection?
5. In fig.6, the authors show that the NANA mutant is impaired in resection but it's not completely abolished as the absence of HLTF. The authors should dissect the other domains of HLTF to see if they can completely abolish the resection.

Minor Comments:

1. In fig.3a the probe labeling is swapped.
2. In fig.5 the authors refer to work from the Nussenzweig lab to support their data which seems to be unpublished. The authors should either provide the manuscript in addition or remove this reference.

May 14, 2024

Manuscript NCOMMS-24-06709-T

Please note we have renamed the manuscript to better highlight the main message.

Original title: Mechanism of DNA break sensing and processing in CRISPR-based genome editing

New title: **HLTF disrupts Cas9-DNA post-cleavage complexes to allow DNA break processing**

ANSWERS TO REVIEWER COMMENTS

We thank the reviewers for their time and effort to evaluate our manuscript and for providing helpful suggestions. Below we provide a point-by-point response to the individual comments.

Reviewer #1 (Remarks to the Author):

The Cejka laboratory has investigated mechanisms that regulate the efficiency of repair of DNA double-strand breaks introduced by Cas9. By biochemical means, the authors have found that the Cas9 breaks are poorly resected by the Mre11-Rad50-pSae2 or Mre11-Rad50-Xrs2-pSae2 complex. Additional experiments help establish that Cas9-introduced breaks are more refractory to end resection enzymes because of end bridging by Cas9. The authors further demonstrate by single-molecule and bulk biochemical analyses that the motor protein HLTF can remove Cas9 from DNA to facilitate resection and hence DSB repair.

The data presented are of high quality and cogently explain why DNA double-strand breaks introduced by Cas9 are repaired more slowly than other breaks, and they also implicate HLTF in the conversion of Cas9-associated breaks into a structure amenable to DNA end processing and repair. This is a very nice study that is suited to Nature Communications pending revisions.

Major Comments:

1. Is there direct evidence that DNA ends stay bridged by Cas9 in the biochemical analyses?

REPLY: The 3-fluorophore experiments in Supplementary Fig. 4d-f demonstrate that the two DNA ends are bridged, under our resection conditions. In particular, the E23 ratio in panel f reflects the colocalization of the two DNA ends, labeled with Alexa750 and Cy5. After DNA cleavage by Cas9, the majority of DNA molecules have a higher E23 ratio than after incubation with HLTF, demonstrating that a large proportion of the substrate is bridged before the action of HLTF. We have added to the legend the following sentence: "**The E21 ratio was used as a measure of Cas9 binding to the DNA while E23 was used as a measure of whether the two DNA ends were bridged.**" The bridged nature of the ends after cleavage by Cas9 is well reported in literature [PMID: 24476820, 29442507].

2. Why aren't degradation products generated by EXO1 detected in Figure 1d.

REPLY: The DNA degradation products are mainly mono- and di-nucleotides, which are not stained effectively by the DNA dye used in the experiment (GelRed - Biotium). We have specified in the figure legend that the DNA was stained with GelRed.

3. Rad5 is the yeast ortholog of human HLTF. It is not clear why it was not tested along with other yeast proteins.

REPLY: We have attempted to prepare yeast Rad5, but failed to do so. Because of the relevance of the manuscript for genome editing in human cells, we focused our attention on the human proteins, in particular on HLTF.

4. The model in Figure 6K suggests that HLTF interacts with 3' overhangs generated by Cas9 activity. Does HLTF requires Cas9 for its recruitment? Have the authors checked if HLTF but not HLTF NANA is recruited to DNA after Cas9 has acted? Do Cas9 and HLTF physically interact, perhaps through HIRAN of the latter?

REPLY: We thank the reviewer for this comment. Our model suggests that the HIRAN domain, thanks to its well described ssDNA binding activity [PMID: 25858588, 26350214], binds the 3'-overhang created by the WT and HA variants of Cas9. To directly address this point, we cleaved DNA by Cas9 HA (creating DNA products with a 3' overhang) and Cas9 DA (no 3' overhang). We then performed DNA binding experiments using the HLTF HIRAN domain (either WT or NANA variants). The best DNA binding was observed using the WT HIRAN domain on top of Cas9 HA, in agreement with our model. Reciprocally, the binding of HIRAN NANA to Cas9 DA-cleaved substrate was severely compromised. We included these data in new Supplementary Fig. 7fg. We do not assume the proteins physically interact, as they evolved independently.

5. Resection of Cas9 breaks occurs upon addition of the yeast helicases Srs2 and Mph1 in ED Figure 3a. This result should be discussed.

REPLY: Since HLTF was much more active in Cas9 removal than Srs2, Mph1 and FANCM (the human orthologue of Mph1) we focused our attention primarily on the mechanism of Cas9 removal by HLTF, also because of the relevance for human genome editing. Our cellular data suggest that HLTF entirely abrogates checkpoint signaling triggered by the Cas9 HA nickase (Fig. 5e), suggesting that HLTF is the main factor acting on Cas9 HA nickase post-cleavage complexes in cells.

6. Is MRX more, or less, efficacious than MRN in cleaving the Cas9 intermediate? The authors should provide quantitation to answer this question.

REPLY: We thank the reviewer for the suggestion. *In vitro* experiments from our laboratory show that yeast MRX is generally more active than human MRN [PMID: 25231868, 27889449], independently of Cas9. The generally lower specific activity is then reflected in a lower capacity to process Cas9-mediated DSBs. We added an experiment (new Fig. 4b and Supplementary Fig. 3e) that shows a comparison of the yeast and human MRE11 complexes to clarify this point.

Minor Comments:

1. Figure 1C: Some resection is seen in the presence of Cas9 (lane 4). Authors should modify the statement "...while DNA cut by Cas9 was not processed".

REPLY: We thank the reviewer. We modified the text as suggested.

2. Have the authors checked if cleavage products generated by Cas12a are a suitable NHEJ substrate?

REPLY: We tested ligation of ends created by Cas12a (Fig. R1). The cleaved ends are not amenable for end-joining, most likely because of Cas12a binding. HLTF is unable to rescue this activity.

Figure R1. Representative in vitro non-homologous end-joining (NHEJ) assay of the plasmid-length DNA substrate (1 kbp) cleaved with HindIII and/or Cas12a. The empty arrowhead indicates the substrate before cleavage (lane 1). The full arrowhead indicates cleaved DNA (0.5 kbp, lanes 2-7). The bracket (Ligation products*) refers to lanes 2-7, and indicates DNA ligation products, along with a fraction of uncleaved DNA (lanes 2 and 5).

3. Figure 4d: degradation products are not visible.

REPLY: The DNA degradation products are constituted mainly by mono- and di-nucleotides, which are not effectively stained by the DNA dye used for the experiment (GelRed - Biotium). We specified in the figure legend that the DNA was stained with GelRed.

4. Have authors tried to perform NHEJ assays in the presence of higher HLTF concentrations (Figure 4f)?

REPLY: The reviewer probably noted the low efficacy of the end-joining reactions after Cas9, compared to ends generated by EcoRV. We believe that the low efficacy is due to Cas9 producing partially non-compatible ends, because the incision points are not precisely defined (see cartoon in Fig. 1a) [PMID: 22745249]. We clarify this point in the revised manuscript. Higher concentrations of HLTF did not increase the efficacy (on the contrary, efficacy was reduced, Fig. R2). We speculate that HLTF can strip the NHEJ machinery from the DNA due to its dsDNA translocase activity, or otherwise non-specifically compete for DNA binding - high HLTF concentrations inhibited also joining of ends generated by EcoRV.

Figure R2. Representative in vitro non-homologous end-joining (NHEJ) assay of the plasmid-length DNA substrate (1 kbp) cleaved with EcoRV and/or Cas9 in the presence of increasing concentration of HLTF. The samples in lanes 16-21 were incubated for 5 min at 80 °C before addition of the NHEJ machinery to remove Cas9 from the end. The empty arrowhead indicates the substrate before cleavage (lane 1, 8 and 15). The full arrowhead indicates cleaved DNA (0.5 kbp, lanes 2-7, 9-14 and 16-21). The bracket (Ligation products*) refers to lanes 2-7, 9-14 and 16-21, and indicates DNA ligation products, along with a fraction of uncleaved DNA (lanes 9 and 16).

5. The NANA mutation in HLTf reduces the resection efficiency by 50% compared to WT HLTf (Figure 6F). Discuss this thoughtfully, as it suggests that another HLTf region is also involved in Cas9 removal. We expressed and purify a HIRAN deletion mutant (1-180 aa) of HLTf. The mutant is completely deficient in multiturnover assay and in the annealing DNA end resection experiments but we note that the branch migration activity of the protein is severely compromised. We nevertheless believe that the main determinant for Cas9 removal is the HIRAN domain, conclusion supported by the dominant negative effect of the HIRAN domain over wild-type HLTf.

REPLY: We thank the reviewer for this comment. In principle, it is possible that other HLTf regions are involved in Cas9 removal, or, alternatively, the point mutations do not completely abrogate the function of the HIRAN domain under our experimental conditions. To address this point, we expressed and purified a HIRAN deletion mutant of HLTf (lacking residues 1-180, Fig. R3ab). The mutant was completely deficient in the annealing DNA end resection experiments (Fig. R4c) and in the multiturnover assays (Fig. R3d), suggesting that it is completely incapable of Cas9 removal. However, we note that the branch migration activity of the truncated protein was reduced about 10-fold (Fig. R3e), possibly due to conformational/allosteric effects resulting from removing a large domain from the polypeptide. We nevertheless believe that the data support that the main initial determinant for Cas9 removal is the HIRAN domain of HLTf, subsequently supported by HLTf motor activity. The conclusion is additionally supported by the by the observed dominant negative effect of the HIRAN domain on top of wild-type HLTf (Fig. 6h-j), and on the reactions requiring ATP (Fig. 4g) demonstrating the requirement for HLTf translocase activity.

Figure R3. **a**, A cartoon representation of the domain structure of the full-length HLTf proteins (Wt and NANA) and the purified HIRAN domain deletion mutant. **b**, Coomassie-stained gel of HLTf WT, HLTf NANA and HLTf ΔHIRAN variants. 1 μg of each protein was loaded on the gel. **c**, Annealing DNA end resection assay of Cas9 breaks by MR-pSae2 in the presence of HLTf WT, HLTf NANA or HLTf ΔHIRAN variants. Top, quantitation. $n \geq 3$; error bar, SEM. Bottom, a representative experiment. DNA resection efficiency was normalized to the sample with 20 nM HLTf WT as 100%. **d**, Quantitation of Cas9 multi-turnover assay in the presence of 0.5 nM Cas9, 1 nM DNA, and increasing concentration of HLTf WT, NANA and ΔHIRAN variants. $n \geq 3$; error bars, SEM. The data for the NANA is reproduced from Fig. 6e. **e**, Quantitation of DNA branch-migration experiments by HLTf WT, NANA and ΔHIRAN variants. $n = 3$; error bars, SEM. The data for HLTf WT and NANA is reproduced from Fig. 6b.

6. Figure 2, C and F: Some of the error bars are compromised; the y-axis needs to be adjusted to fix this.

REPLY: In the particular example of Fig. 2c, the values were normalized to the linear DNA as 100% and, as such, the value for linear DNA does not have error bars. This point was clarified in the label of the y-axis and in the figure legend.

Reviewer #2 (Remarks to the Author):

In this manuscript, Reginato et al. addressed two key questions: Firstly, the Cas9 residing on cleaved DNA conceals the DSB from access by repair factors and delays the repair process, e.g. end process initiated by MRN complex (MRX in yeast). Exposure of loose DSB end activates MRN which endonucleolytically cleaves 5'-terminated DNA strand even with the protein blocks, such as dCas9, bound adjacent to the DNA end. Then backward resection in a 3' to 5' direction by MRN removes the protein blocks and creates the 3' terminals. While the Cas12a releases the PAM-distal end of DNA and initiates MRN-mediated resection immediately. This unique behavior of Cas9, and consequently impact on repair are well known. Secondly, it provides strong in vitro evidence that HLF1 removes DNA bound Cas9 (or H840A) and converts Cas9 or H840A into multi-turnover enzymes. Removal of Cas9 or H840A initiates subsequent DDR and repair. Mechanistically, HLF1 used the HIRAN domain to recognize the free 3' non-target ssDNA released from the Cas9-sgRNA-DNA complex. Therefore, HLF1 only dislocates H840A nickase but not D10A. These findings revealed a new regulator in controlling the target residence of Cas9, in addition to DNA metabolism events including transcription, replication and remodeling. This study not only provides valuable insights into the understanding of DNA damage and repair when Cas9 is used to introduce the breaks, but also be very helpful to the gene editing community, especially for the Prime editor system developed on H840A. Meanwhile I notice a paper (ref 14) that has also demonstrated the HLF1 function in release of H840A, but I am unable to access the original manuscript since this work has not been published yet. Taken together, the data are convincing and support their conclusions. I do have several concerns before the acceptance.

1. In the MR-pSae2 resection assays. To active the MR complex, it requires a 3 to 8-fold excess of pCtIP (p-Sae2) over MR? For Fig1a. the radioactive probe was labeled at a distance about 90nt away to the Cas9 bound end, but in ref50, the distance is 160nt. The probes were designed to move toward the DNA end due to the bound Cas9 on DNA?

REPLY: We used an excess of pSae2/pCtIP because their active form is likely a tetramer [PMID: 25558984, 25580577, 30275497] and not being the active component, we decided to add them in non-limiting amounts. Since the probe used in ref50 was well-tested, we decided to use it for the detection of resection of Cas9 DSBs. Because of PAM positioning, the substrate used for the Cas9 experiments is different than the one used in ref50, and therefore the probe position was closer to the DNA end. Additionally, the probe closer to the end made resection detection easier.

2. In Fig1C, WT Cas9 cutting demonstrated a very weak band showing limited resection by MR-pSae. Because the reaction was performed for a short time, it is possible that more Cas9 would dissociate from cleaved DNA spontaneously and promote the end resection by MRN with longer incubation time (more than 10h)?

REPLY: We thank the reviewer for the comment. To address this point, we incubated the substrate with Cas9 for longer time (Supplementary Fig. 1). Under our conditions, the Cas9-DNA complex is stable for at least 10 hours, as we could not detect any increase in the resection signal during that time. The reported half-life of the Cas9 post-cleavage complex *in vitro* was up to 5.5 hours [PMID: 24476820, 26789497].

3. In Fig2b, the schematic representation of DNA substrate did not be correctly described in the text? Nicked circular 1 indicates the substrate with *Lacl* block placed 0.9kb away from the nick, and Nicked circular 2 for the substrate with *Lacl* block placed adjacent to the nick.

REPLY: We thank the reviewer for the comment. We corrected the text accordingly.

4. In Fig2c. Nicked circular 2, with blocks placed adjacent to nick, could be processed by MR-*pSae*, albeit to a much less extent. The MRN can somehow sense the nick end and initiate resection?

REPLY: The reviewer is correct. The MRX endonuclease activity appears to be partially activated by a nick next to a protein block but to a much lower extent compared to a double-stranded DNA break. We are trying to understand the mechanism of end recognition by MRX, but this is a focus of another study.

5. Fig3A. The resection in the case of *EcoRV*+*Cas9* (lane 5 or 10) is much stronger than that of *EcoRV* cutting along (lane 3 or 8), because the *Cas9* block stimulates the MRX endonuclease activity. Post *EcoRV* cleavage, The *Cas9* bound on one end can stimulate the MRN-mediated resection of DNA on the other side?

REPLY: We thank the reviewer for the comment. We could not find a compelling and all-encompassing explanation for the increase in the signal when *Cas9* was present in the reaction since this effect was reproducibly observed independently of the *Cas9* orientation and probe position (Fig. 3a and Supplementary Fig. 3a). A possibility is that the presence of a protein block at one end could affect resection of the other end (it was speculated that resection occurs on both ends in a coordinated manner, e.g. PMID: 23555316 and 25580577), or that the protein block on one half of the DNA ends could affect the amount of available residual MRX in the reaction, or for other technical reasons.

6. Fig3C. lane4. The band at 30nt indicates the products which MR-*pSae* incises the Ku-bound DNA 30nt away from the end. A weaker band at 60nt suggests another incision point of *Mre11*? Whether it occurs on d*Cas9* bound DNA?

REPLY: The primary cleavage point of the Ku-blocked DNA is at 25-30 bp away from the 5'-end, which is consistent with footprint of the Ku complex [PMID: 29321177, 29321179, 36917982]. A secondary cleavage is observed at 45-50 bp, consistent with ability of the MRE11 complex to introduce multiple tandem incisions past the protein block [PMID: 30819891, 36917982]. A band at a similar location seems to be the primary incision of the d*Cas9*-bound DNA. This could be due to the combination of the size of *Cas9* footprint and its position on the DNA substrate. We also note that the MRE11 complex has a certain sequence preference that could direct the incision in the same region [PMID: 36917982]. The absence of a secondary cleavage of the d*Cas9*-bound DNA could be due to the size limitations of the substrate and the possible presence of a MR molecule at the other DNA end. We updated the cartoon to better reflect the observed DNA cleavage points (updated Fig. 3d).

7. Fig3F. With an excess of *Cas9* (5nM) over DNA (1nM), there is still considerable level of uncut substrate. If these DNA could be ligated, the ligation products would be observed as larger bands on the gels, similar to those present in lane3, in which *EcoRV* cleaved DNA substrates could be ligated to form the large DNA products.

REPLY: We thank the reviewer for this comment, and we realize that the assay was not sufficiently described/explained. The substrate used for the NHEJ experiments is produced by PCR and therefore does not have phosphorylated ends. This setup was necessary to prevent end-to-end ligation that would mask ligation of *Cas9*-generated ends. The uncleaved DNA, present possibly due to a fraction of inactive *Cas9* RNPs that prevent DNA cleavage by active complexes, cannot be ligated due to the absence of

phosphates. We propose that the larger bands observed with *EcoRV* and, to a lesser degree, with Cas9, are due to the Cas9/*EcoRV*-generated ends (phosphorylated) ligating to non-phosphorylated ends *via* a single phosphate group. This point was clarified in the figure legend.

8. Fig4d. HLTF dislocates the Cas9 and exposes the DNA for Exo1 degradation. Whether Exo1 recognize the nicked DNA and initiate resection when HLTF dislocate H840A?

REPLY: We thank the reviewer for this comment. We tested Exo1 degradation of substrates treated with Cas9 nickases HA and DA (new Supplementary Fig. 6a). Exo1 is capable of degrading DNA from the nick generated by Cas9 HA in the presence of HLTF. This was not observed with Cas9 DA.

9. Follow the above question. In Line 152. The authors indicated “Such DNA end sensing capacity prevents spurious cleavage of genomic DNA away from DSB sites”. Based on the results, the MRN would not initiate resection from the nick without a loose DSB end. In other words, the Mre11 would not process the DNA nicks or gaps in genome without a DSB present adjacent to them? Recently, a study (Nature Communications (2023) 14:6265) showed the Mre11 and Exo1 bidirectionally process the replication-derived ssDNA gap and create a DSB when the gap is extended. It suggests the distinct mechanism of MRN resection in mammalian cells.

REPLY: The reviewer is correct, we tested the activity of MRX at gaps for a different project and we observed that only the exonuclease activity is efficiently activated, at least *in vitro* with the yeast complex. In contrast, activation of the endonuclease activation requires the presence of a loose DSB. We specified in the text that the DNA end sensing capacity specifically controls the endonuclease activity of the complex.

10. The Cas12a resides on one end of the DNA break. How does Cas12a dissociate from DNA. The dissociation of Cas12a from DNA requires a specific translocase?

REPLY: Although not proven directly, it is generally hypothesized in the field that one of the function of the MRE11 complex is to remove the protein blocks from DNA ends (such as the Ku complex or stuck translocases). We therefore speculate that Cas12a removal is mediated by the MRE11 complex. To address this point *in vitro*, we tested whether MR-pSae2 was able induce Cas12a multiturnover behavior, which would suggest removal (Fig. R4). Indeed, by comparing the substrate utilization between lanes 3 and 5, an increase of the cleavage capacity of Cas12a can be observed in the presence of MR-pSae2.

Figure R4. Representative Cas12a multiturnover assay. The substrate was incubated with Cas12a for 10 min at 37 °C and subsequently incubated with MR-pSae2 at 30 °C for the indicated time.

Reviewer #3 (Remarks to the Author):

This manuscript by Reginato and colleagues beautifully show that in vitro, Cas9 breaks cannot be resected by the MRE11 complex. They furthermore show that to allow resection, Cas9 is required to be

dissociated by the translocase HLTF which directly removes Cas9 from the broken ends which then allows for further processing. The authors also show that the removal of Cas9 is mediated by the binding of HIRAN domain of HLTF and also the release of 3' end by Cas9. This is an important study, which mechanistically elucidates why Cas9 generated breaks takes time to be repaired. The work has been well designed and provides critical mechanistic insights for the mechanism of Cas9 mediated DSB repair. Below are some comments which need to be addressed before the manuscript is published.

Comments:

1. It is unclear to me why the looped DNA substrate (fig2.d and e) is resected by MR-pSae2. From fig2.b it is clear that probably a double stranded end with the protein block is required for resection. However, the looped substrate wouldn't resemble a double stranded end. How is then the break sensed to initiate resection?

REPLY: We suggest that the activation of the MRX endonuclease activity is dependent on the presence of a loose end (i.e. not a bridged end), rather than the actual chemistry. We also note that the end can be covered by a protein block and still be recognized and processed. As noted above, we do not understand the mechanism behind the end recognition yet. The looped (i.e. hairpin-capped) end is therefore sensed and processed as a DSB, as known from genetic experiments [PMID: 11832209], and as previously demonstrated also *in vitro* [PMID: 29321177].

2. In fig3.c the authors show that using Ku or dCas9 the incision point by MR-pSae2 is dependent on the footprint of the protein block used. Would they also observe similar resection if an active Cas9 were to be used instead of dCas9 on the same substrate?

REPLY: We performed an experiment comparing MR-pSae2 cleavage of a 3'-labeled DNA substrate next to Cas9 WT and dCas9 (Fig. R5). We note that the incision point was the same.

Figure R5. **Top:** A cartoon of the assay shown in C. The positions of the observed DNA incision points are indicated by scissors. Higher transparency is used to indicate lower cleavage efficiency. The red asterisk shows the position of the 32P label. **Bottom:** Representative MR-pSae2 endonuclease assay with a 81 bp 3' radioactively labeled oligonucleotide-based substrate in the presence of catalytically active or dead Cas9. The position of the various cleavage products is indicated by arrowheads (refer to panel a for the position of the cleavage products).

3. In fig.4b the authors show that FANCM also removes Cas9 and promotes resection. This is an interesting observation which was not followed up on. Is the mechanism of FANCM removal of Cas9 similar to the one observed by HLTF. The authors need to parse this out a bit more. Furthermore quantitation of FANCM mediated removal of Cas9 is missing in comparison to HLTF. Also, FANCM removal of Cas9 in combination with MRN-CtIP should also be shown along with MR-pSae2.

REPLY: We thank the reviewer for this comment. As suggested, we compared the activities of HLTF and FANCM quantitatively (new Supplementary Fig. 4d), and it is apparent that FANCM is much less active. Because our cellular data suggest that HLTF is the key motor protein acting on Cas9 HA (Fig. 5e), which is

also supported by upcoming data from the Nussenzweig laboratory, we focused the paper on the analysis of HLTF, which we know is physiologically relevant.

4. I am a bit confused with fig.3c and extended fig3f. In fig3.c the authors show that use of dCas9 can have incision by MR-pSae2. However, in extended fig.3f, the authors show that HLTF addition of HLTF to dCas9 does not induce resection. Is it because HLTF cannot bind to the 3' end generated by active Cas9? Can addition of FANCM to this reaction induce resection?

REPLY: MR-pSae2 only incise DNA near ends. In Supplementary Fig. 3f, an end is not generated (dead Cas9 does not cleave plasmid DNA) and therefore the complex nuclease activity is not activated, regardless of the removal of Cas9 by HLTF or other translocases. In Fig. 3c instead, the oligonucleotide-based substrate contains two DNA ends that activate the complex, which allowed us to monitor the incision points by MR-pSae2, and the effect of Cas9 as a protein block on the incision activity.

In fig.6, the authors show that the NANA mutant is impaired in resection but its not completely abolished as the absence of HLTF. The authors should dissect the other domains of HLTF to see if they can completely abolish the resection.

REPLY: Please see our response to comment 5 of Reviewer #1. We believe that the NANA point mutations may not have fully inactivated the HIRAN domain. In fact, truncation of the HIRAN N-terminal domain entirely abolished HLTF capacity to remove Cas9. Unfortunately, the mutant also showed a 10-fold reduced branch migration activity *per se*. Nevertheless, the experiment demonstrated the unique importance of the HIRAN domain. We note that due to the location and the importance of the SNF2-type ATPase-helicase domain, it would be challenging to produce additional HLTF deletion mutants that would retain their motor activity.

Minor Comments:

1. In fig.3a the probe labeling is swapped.

REPLY: We thank the reviewer for the comment. We corrected the text.

2. In fig.5 the authors refer to work from the Nussenzweig lab to support their data which seems to be unpublished. The authors should either provide the manuscript in addition or remove this reference.

REPLY: The related manuscript is under revision in another journal, and it will take some time before being published. We have opted to remove the statement from the text. However, we can provide the manuscript on request to the editor.

REVIEWERS' COMMENTS

Reviewer #1 (Remarks to the Author):

Authors have done a good job addressing all the comments made previously.

This is a nice study suited to publication in Nature Communications.

Reviewer #2 (Remarks to the Author):

The authors have satisfactorily addressed my questions and I believe the manuscript is ready for publication in the current form.

Reviewer #3 (Remarks to the Author):

The authors have sufficiently addressed my comments and I recommend the manuscript for publication.